# From E-Waste to High-Value Materials: Sustainable Synthesis of Metal, Metal Oxide, and MOF Nanoparticles from Waste Printed Circuit Boards

**DOI:** 10.3390/nano14010069

**Published:** 2023-12-26

**Authors:** Tatiana Pineda-Vásquez, Leidy Rendón-Castrillón, Margarita Ramírez-Carmona, Carlos Ocampo-López

**Affiliations:** 1Centro de Estudios y de Investigación en Biotecnología (CIBIOT), Universidad Pontificia Bolivariana, Circular 1ª No 70-01, Medellín 050031, Colombia; tatiana.pineda@upb.edu.co; 2Centro de Estudios y de Investigación en Biotecnología (CIBIOT), Chemical Engineering Program, Chemical Engineering Faculty, Universidad Pontificia Bolivariana, Circular 1ª No 70-01, Medellín 050031, Colombia; leidy.rendon@upb.edu.co (L.R.-C.); margarita.ramirez@upb.edu.co (M.R.-C.)

**Keywords:** recycling WPCBs, green chemistry, e-waste, bioleaching, photocatalysis, nanoparticles

## Abstract

The exponential growth of electronic waste (e-waste) has raised significant environmental concerns, with projections indicating a surge to 74.7 million metric tons of e-waste generated by 2030. Waste printed circuit boards (WPCBs), constituting approximately 10% of all e-waste, are particularly intriguing due to their high content of valuable metals and rare earth elements. However, the presence of hazardous elements necessitates sustainable recycling strategies. This review explores innovative approaches to sustainable metal nanoparticle synthesis from WPCBs. Efficient metal recovery from WPCBs begins with disassembly and the utilization of advanced equipment for optimal separation. Various pretreatment techniques, including selective leaching and magnetic separation, enhance metal recovery efficiency. Green recovery systems such as biohydrometallurgy offer eco-friendly alternatives, with high selectivity. Converting metal ions into nanoparticles involves concentration and transformation methods like chemical precipitation, electrowinning, and dialysis. These methods are vital for transforming recovered metal ions into valuable nanoparticles, promoting sustainable resource utilization and eco-friendly e-waste recycling. Sustainable green synthesis methods utilizing natural sources, including microorganisms and plants, are discussed, with a focus on their applications in producing well-defined nanoparticles. Nanoparticles derived from WPCBs find valuable applications in drug delivery, microelectronics, antimicrobial materials, environmental remediation, diagnostics, catalysis, agriculture, etc. They contribute to eco-friendly wastewater treatment, photocatalysis, protective coatings, and biomedicine. The important implications of this review lie in its identification of sustainable metal nanoparticle synthesis from WPCBs as a pivotal solution to e-waste environmental concerns, paving the way for eco-friendly recycling practices and the supply of valuable materials for diverse industrial applications.

## 1. Introduction

### 1.1. Significance of Waste Printed Circuit Boards (WPCBs)

E-waste, or electronic waste, is a rising global environmental concern due to the ever-accelerating accumulation of discarded electronic devices. In 2019, the global production of e-waste reached 53.6 million metric tons, and projections indicate a surge to 74.7 million metric tons by the year 2030 [1]. With electronic waste production currently estimated at around 52.2 million tons annually and a growth rate of 5%, the importance of recycling becomes increasingly evident [2].

Electronic waste includes a variety of components, with waste printed circuit boards (WPCBs) representing about 10% of total e-waste. WPCBs are of particular interest due to their high content of valuable metals like copper (Cu) and tin (Sn); precious metals such as silver (Ag), gold (Au), and palladium (Pd) [3]; and rare earth elements such as praseodymium (Pr), neodymium (Nd), lanthanum (La), and cerium (Ce), if compared with mineral ores [4]. The presence of these metals make electronic waste a potential resource for urban metal mining, reducing the need for virgin ore extraction [2]. However, it is crucial to acknowledge that e-waste also harbors hazardous elements like cadmium (Cd), mercury (Hg), nickel (Ni), and lead (Pb). Inadequate handling of these dangerous substances presents significant health and environmental risks [1].

Recycling WPCBs is imperative, for a triad of compelling reasons, namely environmental stewardship, resource conservation, and the promotion of circularity. From an environmental standpoint, the responsible recycling of WPCBs mitigates the hazardous impact associated with improper disposal, preventing the release of toxic elements into the environment. Embracing a circular economy, WPCB recycling promotes the reuse of extracted materials in manufacturing processes, reducing the reliance on virgin resources and minimizing overall waste generation [5].

### 1.2. Sustainable Recycling Methods

In pursuit sustainable recycling, a variety of methods, including mechanical processing, pyrometallurgy, and hydrometallurgy, have been investigated. The use of cyanide, which increases toxicity and environmental concerns, as well as the generation of wastewater and operational costs as a result of the various stages involved in the processes, presents difficulties, which is a major incentive to discover more sustainable chemical treatments [6].

Alternative reagents, such as thiocyanate, have emerged as greener options for gold and silver leaching. Thiocyanate offers lower toxicity, higher chemical stability, and reduced chemical consumption, making it an attractive choice for sustainable metal recovery [7].

When using aqueous solutions and biological metabolites produced by specific microorganisms, biotechnology approaches like biohydrometallurgy add value to these processes. Bioleaching, for instance, is regarded as a green technology for metal recovery because it is inexpensive to install and operate, consumes little energy, produces no toxic waste, and requires little upfront investment [8].

Furthermore, sustainable recycling methods aim for high recycling rates and minimal emissions. They involve separating the layers and components of WPCBs for reprocessing into various applications. One inspiring outcome of these recycling efforts is the synthesis of high-added-value products, including nano and microstructures, from the recovered metals [2].

### 1.3. Nanoparticles from WPCBs

Due to their exceptionally high surface-area-to-volume ratio, nanoparticles, which range in size from 1 to 100 nm [8], exhibit unique physical and chemical properties [9]. Numerous medical and biological applications of nanoparticles exist, including their use in medical diagnosis [9], medicine, biosensing, healthcare, medication delivery [10,11], and coatings for environmental sterilization [12,13].

Energy consumption, process effectiveness, recovery yield, recycling rate, economic viability [14], pollution levels, and a focus on metals recovery are just a few of the constraints that have an impact on the production of nanoparticles [2]. This is especially true when it comes to the manufacturing of nanoparticles from WPCBs. Although some studies have reported promising outcomes, the methodologies proposed have yet to become practically conducive for large-scale industrial implementation [2]. Recent advancements have employed green chemistry as an eco-friendly approach for nanoparticle synthesis, employing non-toxic solvents and environmentally benign techniques [2]. Deeper innovative techniques must be explored for green metal extraction and nanoparticle synthesis in order to develop the idea that WPCBs could serve as a source of nanomaterials.

### 1.4. Review Objectives

This review explores the opportunities for producing metal nanoparticles from electronic scrap, with a primary focus on WPCBs extracted from personal computers (PCs). The composition of WPCBs and the critical role of pretreatment methods in enhancing metal recovery efficiency were explored. Traditional and green recovery systems have been reviewed, shedding light on the shift towards sustainable practices in this field. This study delves into the realm of nanoparticle synthesis, including methods, concentrations, and purification techniques derived from WPCBs, as shown in Figure 1. A notable emphasis was placed on green synthesis, highlighting its eco-friendly advantages. Lastly, the review comments on the industrial applications of nanoparticles sourced from WPCBs, pointing out the promising future of these high-value materials for use in diverse sectors.

## 2. Composition of WPCBs

WPCBs represent around 3–5% of the total weight of electronic waste [15]. According to a significant amount of research, WPCBs are mostly composed of polymers, ceramics, and roughly, up to 50% metals. The content of Fe, Cu, and Al ranges from 5% to 20%; the content of Zn, Sn, Ni, Pb, and Cr ranges from 0.1% to 10%; and the content of Au, Ag, Cd, Ta, Ta, Ge, Ga, In, Mn, Pb, Pd, Ba, Ti, and Bi is less than or equal to 0.1% [15,16,17,18,19]. Table 1 shows the main composition of metals in WPCBs.

WPCBs contain various valuable metals such as copper, silver, gold, and palladium, some of which are found on pins used for conducting electricity and fixing elements [20]. Au could be found in contacts of the cable. Silver is mainly concentrated on the pins (metal foil contacts) of the electronic components, and its highest content was found in the microchips. Other connector pins are made from Cu–Sn, Cu–Zn, or Cu–Sn–Zn alloys, which are covered with thin Ni or Au layers [16], making PCBs economically attractive for recycling.

## 3. Pretreatment of E-Waste

When it comes to recycled electronics, WPCB pretreatment is a crucial step in the waste’s value-adding process [21]. The effectiveness of the recovery process in producing metal nanoparticles as the end product mainly hinges on this initial stage, which often entails desoldering, disassembling, and physical processing techniques, including crushing and grinding [22]. The WPCB is disassembled to identify its recyclable and dangerous elements; at the beginning of the process, this is achieved employing manual techniques, using tools such as screwdrivers, bench drills, pliers, and hammers, among others [23]. As reported in a study conducted by Ref. [24], few users employed advanced separation equipment, i.e., autoclave treatment in which polymer components are separated from metallic components [25].

Innovative approaches that integrate optical sorting, X-ray spectroscopy, and various physico-mechanical methods have been implemented to achieve effective metal pre-concentration for recycling. In this study, the optical sorting of ceramic capacitors (ECs) is detailed, employing a conveyor belt prototype equipped with machine vision techniques, including a convolutional neural network (CNN) written in Python. Electro-pneumatic nozzles then guide the recognized ECs into sorting bins based on directional airflow, with the recovery rate and sorting accuracy serving as evaluation metrics. Additionally, the authors describe the application of ME-XRT (multi-energy X-ray transmission) sorting for super-large ceramic capacitors (SLCCs), outlining the interpretation of element K-edges from the XRT spectrum and the use of a Python code to simulate sorting performance. The article concludes with the ongoing implementation of ME-XRT sorting into a conveyor system, pending X-ray device operation licensing [8].

Subsequently, mechanical processing is used to separate the metals from the non-metals and reduce the size of the particles in order to enrich some metals for later valorization. This process involves crushing, shredding, and grinding the obtained principal components.

Reducing the particle size through size reduction can generally improve the liberation of materials [23,26,27]. WPCBs should be ground to a fine particle size, often below 5 or 10 mm, for maximal separation. This is most commonly achieved with a ball mill and a hammer mill [18,25], a cutting mill [23], or a knife mill [27]. Table 2 shows the central WPCB pretreatment methods.

The enrichment techniques are specialized processes used after particle size reduction to separate the metals from the non-metals. These operations are based on variations in the physical properties of the materials, such as their density and electrical and magnetic properties, among others. Magnetic separators, electrostatic separators, air drum separators, and laser separators have also been employed. The shape, size, and distribution of the particles affect the quality of particle separation and the effectiveness of the approach. Figure 2 shows the effect of the particle size enrichment method on the efficiency of the recovery of different metals from WPCBs [23,27,32,33,34,35,36,37,38,39,40,41,42,43,44,45,46,47].

According to Figure 2, there is a trend indicating that the e-waste particle size required to obtain an efficiency between 90–100% is between 0.75–8 mm, but most processes opt for a fine grind of less than 1 mm.

## 4. Recovery Systems

### 4.1. Froth Flotation

The enrichment of metal fractions could be accomplished using froth flotation. Due to their high hydrophobicity, nonmetallic particles in the flotation system will develop bubbles and float, whereas metal particles will sink without doing so, accomplishing the separation of metal and non-metal components [48].

Froth flotation is usually used in the first stages to continue with the extractive processes, which include chemical extraction activities, pyrolysis, hydrometallurgical treatments, and electrochemical or biotechnological processes, among others [49]. Figure 3 shows a schematic of the froth flotation process.

### 4.2. Hydrometallurgical Processes

Hydrometallurgy is a chemical metallurgy method that performs the separation and extraction of metals based on reaction in an aqueous medium [4].

When compared to pyrometallurgy, hydrometallurgical techniques can be selective for metals at low concentrations, requiring less energy and emitting fewer gases. Due to the complexity and heterogeneity of these wastes, the liquor produced by the hydrometallurgical process of the WPCBs requires further purification stages because the resulting solution contains many metals.

A variety of leaching reagents are commonly used to recover metals from WPCBs; among the most common are the mineral acids H_2_SO_4_, HCl, and HNO_3_; aqua regia [50,51,52]; cyanide; thiourea [53]; and halogens, such as iodine [53,54,55] and thiosulfate. The toxicity, advantages, and disadvantages of these leaching agents are summarized in Table 3.

Cyanidation has been employed for many years, mainly to recover precious metals (Au, Ag). It is widely used because it is an efficient method that is simple to employ. However, it produces environmental and public concern about its use because of its adverse effects, which has led to further research on alternative leachants, such as the thiosulfate and halide systems [55]. The leaching agents used in the hydrometallurgical process are summarized in Table 4, where aqua regia is still the mixture of acids most used in hydrometallurgical processes to obtain Au, Ag and Pd from e-waste. The use of halogens in the extraction processes has shown an efficiency close to 100% for Au; however, the most recent studies have opted for the use of more environmentally friendly, effective solvents such as thiourea and thiosulfates.

### 4.3. Green Recovery Methods

Over the past few decades, green recovery methods have garnered greater attention within this sector due to their reduced capital investment requirements, lower labor requirements, and decreased energy consumption [31].

Therefore, seizing the opportunities presented by bioleaching is imperative. This technology integrates various disciplines, ranging from geology and microbiology to molecular biology and biochemistry, offering prospects for future utilization.

Biohydrometallurgy, an interdisciplinary field merging biology with the hydrometallurgical process, employs selective and environmentally friendly approaches for metal recovery [59]. By utilizing agents like fungi, bacteria [6], algae, or fermentation products, such as enzymes, it transforms specific substrates into soluble salts within an aqueous medium. Microbial reduction facilitates the recovery of soluble heavy metals [60]. The key methods in this process are highlighted below.

#### 4.3.1. Bioleaching

Bioleaching entails the conversion of an insoluble metal into a soluble form, allowing for its recovery from the leachate through the capabilities of microorganisms, which employ various mechanisms. For instance, chemolithoautotrophic bacteria employ the acidolysis and redox mechanisms, organic acid-producing fungi utilize acidolysis and complexation mechanisms, and cyanogenic bacteria employ complexation [8,61,62,63,64], as shown in Figure 4.

During acidolysis, the surface of the metallic compound becomes protonated, primarily through the action of organic acids produced by heterotrophs (such as malic, oxalic, gluconic, acetic, citric, succinic, pyruvic, and formic acids). Additionally, bacterial inorganic acids like H_2_SO_4_ also possess the capability for acidolysis [63].

Redox is an approach that permits the conversion of insoluble metals into corresponding soluble metallic forms [65]. Next, the complexation mechanism, also referred to as the chelation mechanism, involves the formation of complexes between ligands and target metals within a solid matrix. This leads to the creation of cyanides, organic acids, or siderophores, which play a crucial role in mobilizing metals from waste materials. Complexation typically follows acidolysis and is employed for extracting metals like silver, gold, iron, platinum, and palladium [8,65,66].

The most important parameters involved in bioleaching are microorganisms, microbial concentration, pH, size particles, pulp density, temperatures, aeration, and nutrients [8,19,67,68,69].

Numerous studies have provided evidence that a variety of microorganisms, including fungi and bacteria, can produce inorganic or organic acids, as well as cyanide, to facilitate enzymatic oxidation-reduction, proton-promoted mechanisms, and the formation of ligand complexes [8]. This, in turn, aids in the recovery of heavy metals. For instance, microorganisms like *Aspergillus* spp., *Pseudomonas fluorescens*, and *Chromobacterium* spp. species have been shown to be effective in bioleaching gold from e-waste [62,69,70]. *Acidithiobacillus ferrooxidans*, *Acidiphilium acidophilus*, and *Aspergillus niger* result in the bioleaching of Cu, Ni, Pb, and Zn from e-waste [19,29].

In a process resembling industrial gold cyanidation, cyanogenic microorganisms generate the cyanide lixiviant, which subsequently interacts with solid gold to accomplish the leaching procedure. The cyanide lixiviant originates from the secondary metabolite hydrogen cyanide (HCN), which is synthesized from glycine through enzyme HCN synthase. This enzyme, coded by the hcnABC operon in cyanogenic *Chromobacterium violaceum*, is active for brief periods during the early stationary phase of microbial growth [62]. *C. violaceum* is one of the microorganisms most used for the cyanidation of Au and Ag. Some researchers suggest the combined use of *C. violaceum* with *Pseudomonas aeruginosa* and *Pseudomonas fluorescens* to increase the efficiency of precious metal bioleaching [62,71].

Pulp density remains a pivotal element impacting bioleaching efficiency, in which high pulp density (100 g/L) exerts inhibitory effects on the bioleaching process, leading to reduced metal recovery efficiency. Conversely, low pulp density can impede productivity in large-scale applications. Optimal pulp densities of 0.5%, 1%, and 4% have been identified for the effective biorecovery of precious metals from e-waste [65].

#### 4.3.2. Biosorption

Biosorption is a physicochemical process that involves interactions between biological surfaces or components and ions in a solution. It encompasses both living and nonliving biomass, as well as individual organic compounds, for the purpose of removing metals from e-waste through complexation, chelation, coordination, and ion exchange [65,72].

Various types of biomass, including fungi, bacteria, and algae, have been employed as biosorbents for the recovery of heavy metals, precious metals, and rare earth metals from e-waste [8]. For instance, a study investigated the fungus *Aspergillus* spp. as a biosorbent for cadmium removal from e-waste. This fungus displayed the capability to sequester metal ions from aqueous solutions, achieving a recovery rate of 88% for cadmium (at an initial concentration of 100 mg/L) under controlled conditions, with an optimum pH of 4 and a temperature of 30 °C [73].

Another study utilized the dead biomass of *Aspergillus carbonarius*, immobilized on sodium alginate, for the elimination of hexavalent chromium from wastewater. It achieved a maximum hexavalent chromium removal rate of 92.43% at a pH 2.0 for 12 h at 37 °C, employing a 20 g/25 mL adsorbent dosage [74].

Bacterial microbes exhibit physiological diversity and often require metal ions as cofactors, which makes them well-suited for the biosorption of e-waste metals, since they stimulate enzyme production. Research involving various microbial species, including *Pleurotus florida* and *Pseudomonas* spp., demonstrated that the presence of metals, such as copper and iron, from e-waste stimulated the production of laccase, responsible for extracting and recovering metals from WPCBs [75].

In these studies regarding the metal biosorption from WPCBs, the most critical process parameters include the solution’s pH, which strongly influences the speciation of metal ions and the surface polarity of the biosorbent. The cell walls of microorganisms like yeast or microalgae contain different polysaccharides, which exhibit ion exchange properties [76]. As the literature suggests, biosorbents often contain carboxylic, amine, hydroxyl, and thiol groups, and their acid-base properties are highly dependent on pH, modulating the affinity of these functional groups for other metal ions, and therefore, their selectivity for them as well [77].

The pH in the solution affects the specific uptake at a given equilibrium concentration. At a lower pH, biomass adsorbs more metal ions due to electrostatic interactions between the cells. In contrast, a higher pH hinders the access of metal ions to the binding sites [77,78].

Biosorption is typically conducted in continuous systems using packed bed columns, with microorganisms immobilized in porous matrices, as illustrated in a study by Ref. [76]. The use of residual biomass from various industries is recommended due to its abundant availability, ease of acquisition, and cost-effectiveness ratio [8].

For most biosorption processes, the optimal temperature ranges fall between 20 °C and 40 °C because higher temperatures can cause permanent damage to the cells, whether they are alive or not. Adsorption reactions are generally exothermic, and the extent of adsorption increases as the temperature decreases [74,77,78]. Table 5 shows the overview of the primary green recovery process and the microorganisms used in these methods.

## 5. Nanoparticles from WPCBs: Synthesis, Concentration, and Purification

Following the use of recovery methods that transfer metals into a leach solution, it becomes essential to concentrate and transform the metal ions into value-added metallic elements, nanoparticle systems, compounds, or alloys. The most commonly used methods include chemical precipitation [19], the employment of ion exchange resins, the use of activated carbon [72], solvent extraction, electrochemical processes like electrowinning [29], and other non-conventional methods such as diafiltration or dialysis [83,84,85], the freeze-drying of nanoparticles [86], and the use of sol–gel systems, among others.

Solvent extraction is one of the most used techniques; the principal solvents used in this method include organophosphorus derivatives, guanidine derivations, and a mixture of amines–organophosphorus derivatives to obtain gold particles from the e-waste leachate. The concentration of solvent and the pH of the system are fundamental parameters required for good efficiency in the recovery of metals [87].

Numerous techniques, including precipitation, solvent extraction, electrolysis, ionic exchange resins, and barrier separation, have been suggested for recovering metals from leachate. The non-selective nature of the leaching process complicates the problem and raises challenges in the subsequent purification of the metals. Table 6 summarizes the relevant studies concerning the concentration and purification of metal nanoparticles from e-waste.

Dialysis is a method employed to systematically increase the concentration of nanoparticles within suspensions, while preventing aggregation. This process relies on applying osmotic pressure to a nanoparticle suspension confined within a dialysis bag. As a result of this osmotic pressure, water molecules move from inside the bag towards the external dialysis solution until equilibrium is established on both sides of the dialyzing membrane. This technique is widely implemented, mainly in the pharmaceutical industry, and it has been attracting interest for use in the concentration of metallics, such as gold and rare earth elements, from electronic waste [83,84,85,86].

Electrowinning has demonstrated its effectiveness as a straightforward method for the recovery of certain metals, notably gold and copper, from WPCBs. This method is typically applied to solutions with high copper concentrations (ranging from 25 to 60 g/L) obtained through a series of steps, including leaching and solvent extraction. Conventional electrowinning processes typically involve low to medium temperatures (30–60 °C) and low current densities (below 430 A/m^2^). In electrowinning, pure copper plates are utilized as cathodes [29], while Pb or mixed metal oxide-coated Ti inert plates are employed as anodes [89].

## 6. Green Synthesis of Nanoparticles from WPCBs

In the process of nanomaterial synthesis, the generation of substantial chemical waste has raised massive environmental pollution concerns. While conventional techniques to mitigate the release of impurities, including chemicals and heavy metals, into the environment have been improved, opportunities for improvement in their efficiency, especially in dealing with smaller particles, remain [8]. As a result, ensuring the sustainability of synthesis methods becomes imperative, not only to curtail waste production, but also to optimize the efficiency of both nanomaterial production and waste treatment processes [90].

Microorganisms, plants, animals, and their derivatives, such as amino acids, enzymes, and organic acids, serve as natural sources for facilitating the green synthesis of nanoparticles. Green synthesis stands as a highly promising approach due to its versatility across various applications, offering numerous advantages over conventional methods [25].

The top-down or destructive method involves reducing bulk materials to nanoscale particles. Common techniques used for nanoparticle synthesis include mechanical milling, nanolithography, laser ablation, sputtering, and thermal decomposition. Unlike the bottom-up method, top-down synthesis is a slower and more expensive process [7], and it may not be suitable for large-scale production [91]. The conventional bottom-up method uses highly toxic substances, such as NaBH_4_ and N_2_H_4_, as reducing agents [7], along with extreme temperatures [92] and specific pH conditions [15], and often requires high energy consumption. Few studies report the synthesis of nanoparticles via green routes using WPCBs as a substrate. Most studies use metallic salts, mainly nitrates and chlorides, as precursors. A summary of the most relevant green routes to produce nanoparticles from WPCBs is shown in Figure 5.

### 6.1. Green Organic Acids

The size and well-defined shapes of the nanoparticles depend, to a large extent, on the reducing agent and the action of a stabilizer in the synthesis of the solution phase.

The Turkevich method is a straightforward and reliable approach for synthesizing spherical particles ranging from 10 nm to 30 nm, using WPCBs as a substrate for Ag and Au [7]. The initial concentration of Ag was reported as 15.7 g·L^−1^. In this method, citrates are used as reducing agents [93,94].

A stable suspension of gold nanoparticles was achieved by utilizing a combination of sodium citrate and ascorbic acid as reducing agents, along with a stabilizing agent (polyvinylpyrrolidone—PVP), within a temperature range of 25–65 °C and a pH range of 2.5–4.0. This unique combination of reductive agents and a polymeric stabilizer enabled the production of pure gold metallic nanoparticles, featuring well-defined spheroidal and triangular shapes with a size distribution between 5–20 nm [4].

Ascorbic acid is commonly employed in the synthesis of nanorods. It serves the dual purpose of both reducing and stabilizing during the fabrication of copper nanoparticles (CuNPs) [95,96], silver nanoparticles (AgNPs) [7], and gold nanoparticles (AuNPs) [97]. Furthermore, it acts as an antioxidant, effectively reducing oxygen free radicals and metal ions. As a result, it is considered as a non-toxic reagent which can influence the growth, aggregation, and interaction of the synthesized particles with the external environment [95].

Glucose is another agent used to obtain NPs from WPCBs, and its effect is potentiated by ascorbic acid. In a study on the synthesis of CuNPs, this technique was used. The nanoparticles obtained exhibited round shapes, with sizes between 10–50 nm [98].

### 6.2. Plants

The use of plant leaf extract facilitates the biogenic reduction of metal ions into base metals. This process is characterized by its swiftness, as it occurs rapidly, and its ease of execution at room temperature and pressure. Furthermore, it can be conveniently scaled up. Synthesis mediated by plant extracts is environmentally friendly, and most methods used water [99] and alcohols, such as methanol and ethanol [100], to extract the main components from the plants. Plant extracts, which comprise bioactive compounds like alkaloids, phenolic acids, polyphenols, proteins, sugars, and terpenoids, are thought to play a crucial role in initially reducing metallic ions and subsequently stabilizing them [101]. For example, *Cathantharaus roseus* plant leaf extract was used to generate copper oxide nanoparticles from WPCB, as it was indicated that the amide groups present in the proteins and enzymes of leaf extract are responsible for the oxidation–reduction process, and that amine groups of the leaf extract act as capping agents. The particle size ranges were from around 5 nm–10 nm, exhibiting a polycrystalline nature [25].

In a ratio of 1:20 (solid/liquid), an aqueous extract of olive tree leaves was used as a source of polyphenols for the reduction of Cu, Cr, and Sn metals in WPCBS acidic leachates as a greener alternative for the recovery of valuable metals [99].

An aqueous extract of *Prosopis juliflora* leaves obtained by sonication, containing a pool of piperidine alkaloids, was used in equal proportions with PCB-derived copper nitrate at 80 °C to obtain copper oxide nanoparticles (gCON) for the oxidation–reduction process. The outcomes of optical, structural, and morphological analyses verified the existence of monocrystalline, spherical copper oxide nanoparticles, demonstrating an average size of 11 nm [102].

The leaf extract of *Cassia auriculata* was used as a reducing, as well as capping, agent in the synthesis of copper nanoparticles. The findings from this investigation have substantiated the potential of *Cassia auriculata* leaf extract for the recovery of copper from printed circuit boards, yielding nanoparticles with a size range of 50–100 nm. The CuNPs exhibited a prominent peak at 300 nm in UV–Vis spectra [103].

Aqueous extracts derived from aloe vera and geranium (*Pelargonium graveolens*) were employed to reduce copper ions present in the leaching solution, a product of copper shale (Kupferschiefier) bioleaching via chemolithotrophic bacteria, including *Acidithiobacillus ferrooxidans*. The utilization of *Pelargonium graveolens* extract resulted in the formation of copper nanoparticles (CuNPs) with a spherical shape and a size distribution ranging between 100 and 300 nm [104].

### 6.3. Bacteria

The bacterial synthesis of nanoparticles can occur through extracellular and intracellular processes, with various metals being reported for nanoparticle formation using different bacterial components such as biomass, supernatant, cell-free extracts, and their derivatives. Extracellular synthesis is generally preferred due to the ease with which nanoparticles can be recovered [64,105].

Cytochromes, peptides, cellular enzymes like nitrate reductase, and reducing cofactors all play crucial roles in nanoparticle synthesis within bacteria. Organic materials released by bacteria serve as natural capping and stabilizing agents for metal nanoparticles, preventing aggregation and ensuring long-term stability [106]. Consequently, bacteria are recognized as potential biofactories for synthesizing a wide range of nanoparticles, including gold, silver, platinum, copper, nickel, iron, palladium, titanium, titanium dioxide, magnetite, cadmium sulfide, etc. Given the toxicity of many metal ions to bacteria, the bioreduction of ions or the formation of water-insoluble complexes represents a defense mechanism developed by bacteria to mitigate this toxicity [107].

Intracellular synthesis has been observed, as demonstrated in CuNP synthesis by a bacterium isolated from the marine sponge *Hymeniacidon heliophila*. This bacterium exhibited an affinity for crucial metals released from waste printed circuit boards (WPCBs). Notably, at 30 °C, the bacteria secreted substances beneficial for copper bioleaching, while at 40 °C, metallic nanoparticles formed within the cells. This mechanism is believed to neutralize heavy metal toxicity by reducing the ionic force of metals, encapsulating metallic nanoparticles within the vesicles for later release. This transition from metal ions to non-toxic forms is considered to be a survival mechanism in contaminated environments [27].

Over a three-day period, *Cupriavidus metallidurans* and *Delftia acidovorans* synthesized gold nanoparticles from e-waste, with diameters ranging from 22 to 33 nm. *C. metallidurans* appears to counter gold toxicity through Au-regulated gene expression and the potential methylation of Au complexes, leading to the energy-dependent cellular accumulation of gold nanoparticles. In contrast, *D. acidovorans* employs a distinct gold precipitation mechanism involving the secretion of a secondary metabolite called delftibactin to protect itself from the toxic nature of Au^3+^. In this case, gold nanoparticles are precipitated in the extracellular medium [108].

Finally, the management of waste generated in the plant leaf extract is mainly grouped into the following types: thermochemical treatments, such as combustion, gasification, and pyrolysis; biochemical treatments to obtain, for example bioethanol; drying methods; and the condensation of active components [109].

### 6.4. Fungi

Fungi play a crucial role in biogeochemical processes, capable of both dissolving and immobilizing metals. They have gained attention for their potential for synthesizing nanoparticles as part of biotechnological metal recycling from e-waste due to their remarkable resilience to high concentrations of heavy metals [72]. In contrast, bacterial fermentation processes often entail multiple steps to obtain a clear colloidal, including filtration, solvent extraction, and the use of sophisticated apparatuses that considerably increase the investment costs related to equipment [4].

The biosynthesis of metal nanoparticles using fungal cells can employ one of two mechanisms: (i) intracellular or (ii) extracellular synthesis routes. In intracellular synthesis, nanoparticles are formed and localized within the cytoplasm, cell wall, or cell membrane [110]. Initially, the metal ions, which serve as nanoparticle precursors, interact with oppositely charged cell surface components, where they can be simultaneously reduced to form nanoparticles, while remaining attached to the cell surface. These nanoparticles may then migrate to the cell membrane or cytoplasm. Alternatively, the ions may be internalized through active or passive transport mechanisms and subsequently reduced by intracellular reducing agents [111]. In the extracellular synthesis routes, fungal proteins, enzymes, cofactors, and metabolites like organic acids (e.g., citric acid, oxalic acid) play vital roles in the organism’s survival and contribute to the reduction of metal ions into nanoparticulate forms. Figure 6 provides a schematic representation illustrating the interaction between metals and the fungal cell surface.

Numerous groups of fungi are employed for the synthesis of nanoparticles from WPCBs, with *Fusarium oxysporum* emerging as a prominent organism capable of producing a variety of manufactured nanoparticles. The synthesis of silver nanoparticles (AgNPs) by *F. oxysporum* is primarily dependent on a reductase associated with NADH [4,112].

A mixture of *Fusarium oxysporum* and *Bacillus cereus* from e-waste synthesized Cu, Zn, Cd, and Au nanoparticles. The nanoparticles varied in size, but most of them possessed a diameter of less than 10 nm. They exhibited different shapes, including circular, triangular, and complex [113].

*Trichoderma* spp., *Aspergillus* spp., *Penicillium* spp., and *Verticillium* spp. are additional fungi that offer numerous advantages. These include diverse survival strategies, ease of biomass handling, the capacity to multiply using straightforward culture media, tolerance to high metal concentrations, and enhanced productivity in terms of nanoparticle production from WPCBs [114].

New research could be carried out, specifically using WPCBs as precursors and employing this fungal route for its multiple advantages in the synthesis of nanoparticles.

### 6.5. Algae

Algae represent another species known for its role in the bioreduction of metals, facilitating the production of gold and silver nanoparticles. This capability arises from the presence of specific functional groups, such as carboxyl, on their cell walls [8], along with the secretion of a polysaccharide called fucoidan, which aids in the intracellular synthesis of gold nanoparticles [97]. Table 7 summarizes the relevant studies regarding nanoparticle reduction via the use of algae.

*Chlorella vulgaris*, a unicellular microalga typically found in carbonate and bicarbonate-rich lakes, is widely recognized for its effectiveness in bioreducing metal ions. This is primarily attributed to the production and secretion of cellular reductases by the microalgal cells into their growth medium. These enzymes demonstrate remarkable efficiency in reducing silver ions in order to form spherical silver nanoparticles (SNPs). Depending on the pH conditions, this process results in monodispersed SNPs at low and neutral pH levels and nanorod structures in alkaline conditions [117].

Other algal species, including *Rhizoclonium hieroglyphicum*, *Lyngbya majuscule*, and *Spirulina subsalsa*, have also been identified for their capacity to visibly indicate gold reduction, specifically from Au(III) to Au(0), at both the intra- and extracellular levels. This is evidenced by a color change in their biomass to purple [116].

Green synthesis is presented as a sustainable alternative to traditional methods for the valorization of WPCBs. Within the green techniques mentioned here, some possess important aspects that are presented here as advantages; for example, obtaining nanoparticles using organic acids reduces synthesis times; however, in general, the stability of the nanoparticles obtained, along with their biocompatibility, is reduced. On the other hand, the use of microorganisms, such as bacteria and fungi, provide nanoparticles with defined and stable formats; however, the production cycles are broader, and there are limitations, including their inability to operate at high pulp densities, thus limiting their potential profitability. The use of plant extracts generates stable nanoparticles, but studies report the proportion of the plant extract and the chemical solution as the primary factor affecting the size of the NPs, along with their stability. Impurities of the extracts, limitations in process engineering, and operation stability are considered significant issues. Algae have demonstrated efficiency in the synthesis and stability in regards to the nanoparticles generated, but they present limitations concerning variability in the synthesis processes.

Finally, most of the waste generated in green metal extraction processes, as well as that derived from the synthesis of NPs, can be treated, or valorized, using three major processes, i.e., thermochemical, biological, and mechanical. The thermochemical type includes pyrolysis, gasification, and combustion processes, among others. The biological processes include fermentation, digestion, and the production of enzymes, and finally, the mechanical processes refer to operations such as drying, grinding, and pelletizing, in which added value can be provided to the waste generated in the green metal extraction processes. However, thermochemical processes remain the most required, due to their reduction of waste volumes and their recovery of molten metals [118].

## 7. Industrial Applications of Nanoparticles Based on WPCBs

The nanoscale, characterized by an extensive surface-area-to-volume ratio, offers a diverse range of applications across various fields, including drug delivery, microelectronics, antimicrobial materials, environmental solutions, diagnostics, catalysis, agriculture, biomimicry, nanosensing, antibacterial and antimicrobial products, and textile industry applications. The nanometer scale, which is one billionth of a meter, defines this realm. Nanoparticles possess distinctive structural and property characteristics, driven by their smaller size and the substantial surface-area-to-volume ratio, making them highly sought after for their exceptional properties [119].

Due to the characteristics of nanomaterials, several industrial applications can be developed using nanoparticles obtained from electronic waste as substrates, the most important of which include the following.

### 7.1. Degradation of Dyes

Recently, the use of metallic nanoparticles for the degradation of toxic dyes and persistent substances has garnered significant attention, primarily due to their catalytic activity [120]. Numerous studies have investigated the degradation of various dyes, including methylene blue [102,121], auramine O, thymol blue, rhodamine B dye [96], reactive blue 4 (RB4) [122], and methyl orange, achieving degradation levels of up to 90–100% through the use of silver, copper, iron, and other nanoparticles. Importantly, the dye degradation process involving nanoparticles is rapid and does not involve the use of hazardous chemicals associated with conventional biological and chemical wastewater treatment processes.

A study evaluated the use of CuNPs derived from waste printed circuit boards (WPCBs) through an environmentally friendly process for the degradation of recalcitrant amine surfactants via the modified Fenton reaction. The spherical nanoparticles exhibited an average size distribution of 3.2 nm, leading to a total carbon degradation of 57% [95].

In another case, the removal of methylene blue from an aqueous solution was achieved using CuNPs with an average size of 11 nm. The selective leaching process of WPCBs, employing ammonium chloride and ammonia buffer solutions, was reduced with the assistance of *Catharanthus roseus* plant leaf extract for CuNP generation. Under optimal conditions, involving 20 mg of CuNPs and an original adsorbate concentration of 10 mg/L at an initial solution pH of 8.0, a remarkable 91.6% removal of the adsorbate was attained [102].

### 7.2. Photocatalysis

Photocatalysis is becoming increasingly prominent in the field of pollutant degradation and the inactivation of microorganisms. An ideal photocatalyst should exhibit stability, affordability, non-toxicity, and notably, high photoactivity [123].

Photocatalytic activity is a commonly investigated function of metallic and semiconductor nanoparticles. One of the most studied forms of heterogeneous photocatalysts is titanium dioxide (TiO_2_), in particular, the anatase crystalline form, for the treatment of biological and chemical pollutants in indoor environments [124]. This form is activated by ultraviolet radiation to carry out oxidation–reduction reactions that act on the surface of the microorganism, achieving its inactivation. However, recently, new nanoparticle-modified TiO_2_ photocatalysts that can use visible light as an activator have been investigated, thus extending the applicability of this technology [26,125,126,127].

WPCBs are an essential source of nanoparticles; several reports indicate that the synthesis of nanoparticles has been shown to have a considerable effect on photocatalyst activation for the degradation of volatile pollutants and pigments and the inhibition of bacteria and fungi.

Copper and its oxides sourced from WPCBs have proven to be valuable photocatalysts for the photodegradation of organic pollutants. Their appeal lies in their cost-effectiveness, high optical absorption, and their ideal optical band gap that aligns with visible-driven photocatalytic activity [96]. These copper-based nanoparticles are well-suited for absorbing visible light and generating electron-hole pairs, facilitating chemical reactions with organic pollutants. The synthesized particles exhibit a morphological variety, ranging from spheres to distorted spheres, with average particle sizes of 460 nm for Cu and 50 nm for CuO NPs, respectively. The system displays a significant capability to inhibit *E. coli* and *B. cereus* bacteria, with Cu NPs showing the highest inhibition zone of 21.2 mm against *E. coli* and the Cu/CuO blend showing 16.7 mm against *B. cereus* bacteria, *F. proliferatum*, and *P. verrucosum* fungi. Additionally, the Cu/CuO blend demonstrated remarkable photocatalytic activity in the degradation of Rhodamine B dye under visible light irradiation, achieving a 96% degradation rate within 120 min [96].

In a separate application, a straightforward method was proposed for recovering tin from WPCBs, yielding tin oxide nanostructured powders. WPCBs were subjected to leaching with the disodium salt of the ethylenediaminetetraacetic acid (Na_2_-EDTA) chelating agent. The SnO_2_ NPs obtained, with sizes ranging from 8 to 12 nm, exhibited excellent photocatalytic efficiency in degrading methylene blue (MB) dye under ultraviolet (UV) light, achieving a 90% removal rate after 180 min and displaying good reusability over five consecutive cycles [128].

In the context of pesticide photodegradation, a magnetic metal–organic framework (MOF) incorporating copper recovered from WPCBs was synthesized. This material, known as TiO_2_/mag-MOF(Cu), demonstrated a high level of efficiency in the photodegradation of two prominent organophosphorus pesticides: malathion (MTN) and diazinon (DZN). The composite exhibited strong performance under the following conditions: a pH of 7, a visible light intensity of 75 mW/cm^2^, and a reaction time of 45 min. Superoxide radicals (O_2_^•−^) played a pivotal role in the degradation process, with over 83% and 85% mineralization achieved for MTN and DZN, respectively, as confirmed by total organic carbon (TOC) analysis. This composite looks to be a promising catalyst for the photodegradation of MTN and DZN in aqueous solutions [129].

### 7.3. Coatings

A coating refers to a protective layer of material applied to a substrate for the purpose of shielding it from environmental factors such as corrosion, abrasion, microbiological colonization, and weathering.

One significant area requiring the use of coatings is in the prevention of marine biofouling, where aquatic organisms rapidly colonize ship hulls, leading to substantial economic and ecological repercussions. Consequently, the development of safe coatings, particularly antifoulant coatings, is of paramount importance. Notably, copper and iron nanoparticles obtained from e-waste have been incorporated into paints to safeguard iron surfaces [105].

In textile applications, nanoparticles play a role in providing antimicrobial properties. For instance, AgNPs-chitosan-coated cotton fabric, using AgNPs sourced from e-waste via both chemical and environmentally friendly methods while maintaining pH within the range of 7 to 12, is employed to achieve antimicrobial effects [130].

### 7.4. Biomedical 

Several studies have detailed the synthesis of copper nanoparticles (Cu-NPs) from e-waste using chemical reduction, resulting in crystallite sizes of approximately 27.6 nm. These Cu-NPs find application in various biomedical fields, including their use in drug delivery systems, in the detection of dopamine and amino acids, as well as for antimicrobial activities [131].

In a study conducted by Ref. [17], Cu-NPs were synthesized from a computer RAM sample, featuring an average size of 7 nm. These nanoparticles exhibited potent antimicrobial properties against *E. coli*, *P. aeruginosa*, and *S. aureus*. Remarkably, the production costs of these Cu-NPs were six times lower than those of commercially available alternatives.

Furthermore, spherical selenium nanoparticles (Se-NPs), with sizes ranging from 50 to 500 nm, were synthesized from e-waste and employed as nano-antifungal agents. Their antifungal efficacy was evaluated against *Candida albicans* and *Aspergillus niger* using the disk diffusion method, demonstrating significant antifungal activity when compared to fluconazole [132].

Nanoparticles must be non-toxic and biocompatible to be successfully used in biomedical applications. Due to their small size, nanoparticles can easily enter the body and reach very sensitive organs by different pathways [133].

Nanoparticles have found wide biomedical applications due to their physicochemical and behavioral uniqueness. However, concerns about their toxic effects on the biological system are now attracting the attention of the global health community. This emphasizes the importance of studying and understanding their effects based on the cellular and molecular mechanisms by which they cause harm. Some identified toxic mechanisms include the induction of free radicals (ROS), cytotoxicity to cells, and genotoxic and neurotoxic effects. The toxic effect depends on various factors, such as the type of nanoparticles, as well as their size, surface area, shape, surface chemistry, surface charge, aspect ratio, surface coating, crystallinity, dissolution, and agglomeration [134].

Nanotoxicological studies are planned to determine to what extent these properties could constitute an attack on the atmosphere, animals, and humans. For example, nanoparticles are known to cause damage to the central nervous, circulatory, respiratory, and cardiovascular systems [135], although this damage depends largely on the type of application and how the nanoparticles are synthesized.

## 8. Conclusions and Future Perspectives

It was found that enhancing metal recovery efficiency in electronic waste, especially in WPCBs, relies on processes such as disassembly and manual separation, as well as advanced equipment, like autoclaves. Fine grinding is recommended for optimal separation, utilizing equipment like ball, hammer, cutting, and knife mills. Various pretreatment techniques, including selective leaching, magnetic separation, and hydrometallurgy, offer methods for metal recovery. Enrichment methods focus on achieving 90–100% recovery efficiency for WPCBs particles (0.75–8 mm).

It is important to establish an organized collecting system. As many cities and countries lack this type of regulation, the creation of policies that allow for the optimization of the collection of this type of waste, as well as for its correct and proper classification, is suggested.

Effective electronic waste recycling requires a critical recovery system. Green recovery methods like biohydrometallurgy emerge as sustainable and innovative strategies to recover valuable metals from PCBs. Bioleaching and biosorption are eco-friendly approaches that feature lower resource requirements and high selectivity.

Future research endeavors could combine the strengths of these two methods to further enhance the leaching effect, applying the knowledge that has been gained in the processing of other types of waste specifically to the processing of WPCBs.

Recovering metal nanoparticles from WPCBs involves concentrating and converting metal ions before the synthesis. Other methods, including chemical precipitation, the use of ion exchange resins, and solvent extraction, are added to conventional methods, such as electrowinning and dialysis, for the concentration of metals once they are leached from the solid matrix.

These methods are crucial for transforming recovered metal ions into valuable elements and nanoparticles, promoting sustainable resource utilization and eco-friendly WPCBs recycling.

Addressing chemical waste generation during conventional synthesis is essential. Sustainable green synthesis methods, utilizing natural resources like microorganisms, plants, and their products, offer eco-friendly alternatives with broad applications. Natural reducing agents like ascorbic acid, glucose, and plant extracts produce well-defined nanoparticles, preventing aggregation. Bacterial, fungal, and algal synthesis routes contribute to eco-friendly nanoparticle production, with algae showing unique bioreduction capabilities.

In all these cases, it is important to motivate researchers and industrialists to establish techniques to minimize the waste generated in these eco-friendly methods, including the use of inactivation techniques and subsequent adaptation to obtain inorganic ashes that could be used in construction systems and paving.

Nanoparticles derived from electronic waste, especially WPCBs, find valuable applications across sectors. These applications include their use in drug delivery, microelectronics, antimicrobial materials, environmental remediation, diagnostics, catalysis, agriculture, and more. WPCBs nanoparticles efficiently degrade toxic dyes and recalcitrant substances, providing an eco-friendly alternative to conventional wastewater treatment. 

Nanoparticles also play a role in photocatalysis for organic pollutant degradation in gas and liquid phases and in the harnessing of solar energy for environmental cleanup. They are used in protective coatings, for antifouling solutions, and in antimicrobial textiles. In biomedicine, copper nanoparticles have applications in drug delivery, drug detection, and antimicrobial activities, while selenium nanoparticles exhibit antifungal properties.

Sustainable metal nanoparticle synthesis from electronic waste, particularly WPCBs, reveals promising prospects for addressing environmental challenges and advancing technology across various sectors.

## Figures and Tables

**Figure 1 nanomaterials-14-00069-f001:**
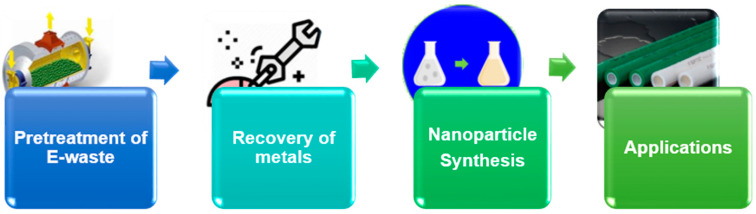
Stages for the recovery and valorization of metals from e-waste.

**Figure 2 nanomaterials-14-00069-f002:**
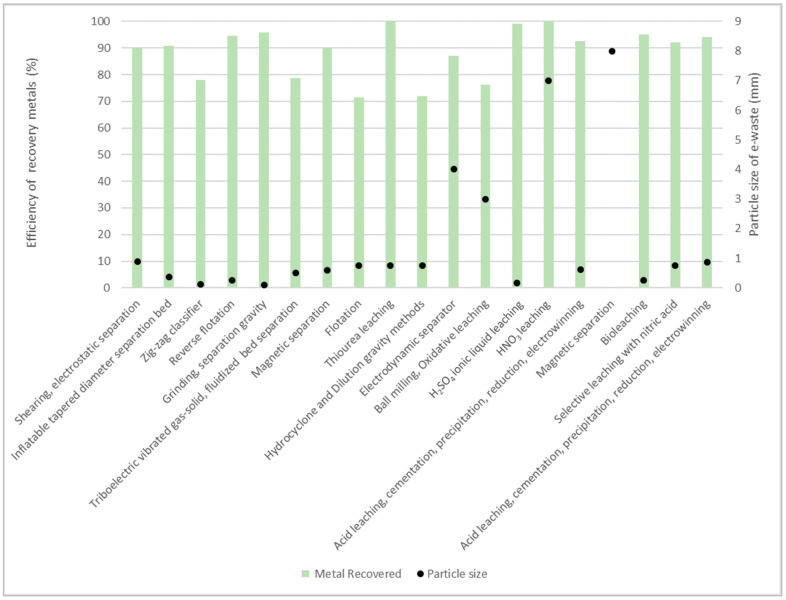
Effect of particle size on the recovery efficiency of precious metals [23,27,32,33,34,35,36,37,38,39,40,41,42,43,44,45,46,47].

**Figure 3 nanomaterials-14-00069-f003:**
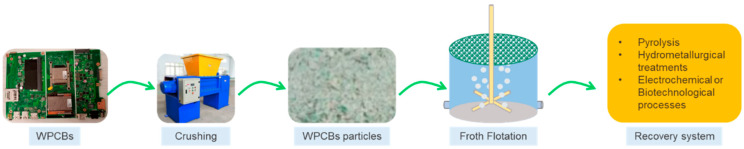
Schematic of the e-waste froth flotation process.

**Figure 4 nanomaterials-14-00069-f004:**
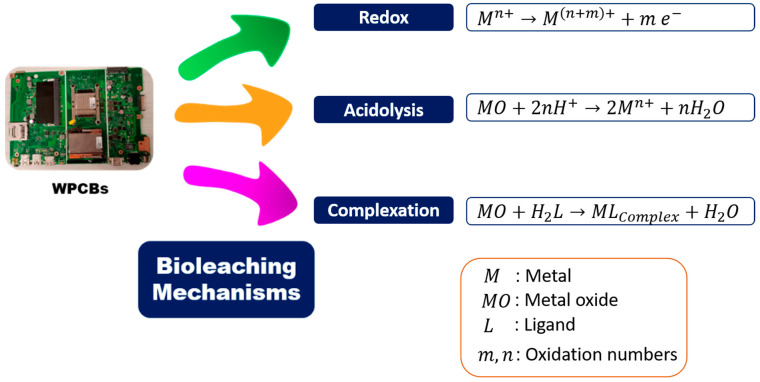
Bioleaching mechanisms applied to WPCBs.

**Figure 5 nanomaterials-14-00069-f005:**
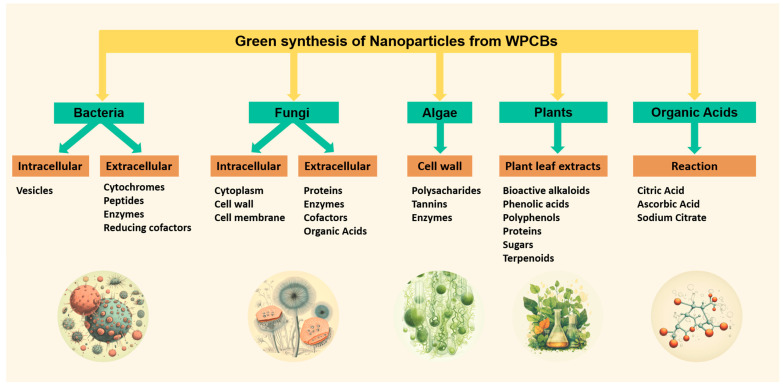
Relevant green routes for producing nanoparticles from WPCBs.

**Figure 6 nanomaterials-14-00069-f006:**
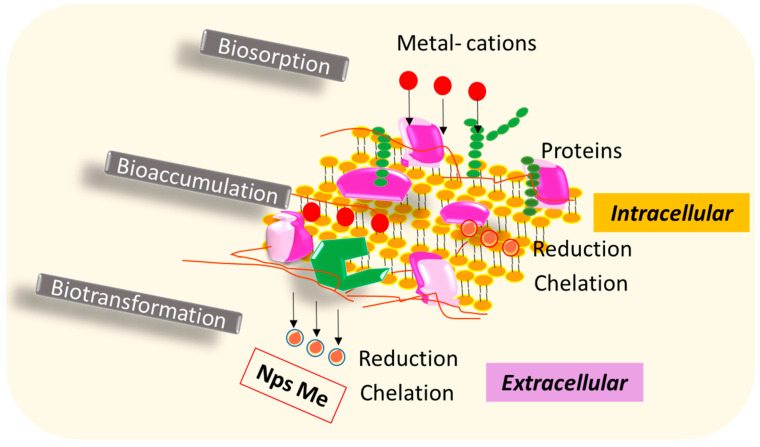
Biosynthesis of metal nanoparticles using fungal cells from e-waste.

**Table 1 nanomaterials-14-00069-t001:** Typical composition of primary metals found in WPCBs, according to literature.

Classification	Percent (%)	Element
Major elements	5–20	Fe, Cu, Al
Intermediate elements	0.1–10	Zn, Sn, Ni, Pb, Cr
Minor elements	<0.1	Ag, Au, Ba, Bi, Cd, Ga, Ge, In, Mn, Pb, Pd, Ta, Ti

**Table 2 nanomaterials-14-00069-t002:** Central WPCB pretreatment methods and the metal nanoparticles obtained.

Type of Pretreatment of E-Waste	Recovery Method	Metal	Nanoparticles	Reference
Autoclave treatment: polymer separation Ball mill–hammer mill	Selective leaching: ammonia	Cu	CuO: 5 nm	[25]
Crushing in a crystal agate mortar, 200 mesh	Selective leaching with nitric acid	Ag	8–30 nm	[28]
Cutting mill: 0.075–1 mm	*Acidithiobacillus ferrooxidans*, *Leptospirillum ferrooxidans*, and *Acidithiobacillus thiooxidans*	Cu	—	[29]
Shredding in a cutting mill equipped with 8 mm mesh	Magnetic separation concentrated by magnetic separation Hydrometallurgical HCl (ranging 1 M to 6 M) and H_2_O_2_ (ranging 4 M to 8.8 M)	Au	AuNP: 5–20 nm	[23]
Cutting: surface areas 100 to 1600 mm^2^	Floating in vibrated fluid ascorbic acid, ß-cyclodextrin, and 80 °C and 700 rpm	Cu	Cu: 7 nm	[17]
Hammer and cutting mill: 0.075–1 mm	Bioleaching: pure culture of *A. ferrooxidans* and citric acid	Cu Zn Pb Ni	—	[18]
Grinding	Thermal concentrated precipitation: conventional, microwave, and ultrasound SnO_2_: metastannic acid AgNp: HCl-ammonia	Sn Ag	SnO_2_: 15 nm AgNp: 0.7 nm and 200 nm	[30]
Griding in a knife mill and crushing in a hammer mill with a 2 mm grid	Bacteria obtained from the marine sponge *Hymeniacidon heliophila* used in selective bioleaching	Cu	Cu: 1 nm	[27]
Cutting mill 0.8–0.4 mm	Bioleaching: pure culture of *A. ferrooxidans* and citric acid	Cu Zn Al	—	[31]

**Table 3 nanomaterials-14-00069-t003:** Toxicity, advantages, and disadvantages of leaching agents used in the hydrometallurgical process.

Leaching Agent	Toxicity	Advantages	Disadvantages
Cyanide	Very High	High dissolution rate	Environmental harmful
Aqua regia	High	High dissolution rate	No feasible large-scale applications are available
Bromide and Iodide	Low	High dissolution rate	High cost of reagents
Thiosulfate	Medium	Economical reagents	Detoxification costs High reagent consumption
Thiourea	High	Proven technology High dissolution rate and speed	Dissolution of heavy metals in addition to gold Carcinogenic compound
H_2_SO_4_	Medium	High extraction rates for easily dissolved species More economical than organic acids Higher current availability compared to organic acids	Environmentally very harmful Requires constant integration of the input into the circuit More complex pH control [56]
HCl	Medium	Second most used leaching agent in metal extraction High solubility Low cost and high availability [57]	Environmentally very harmful Requires constant integration of the input into the circuit More complex pH control [56]
HNO_3_	Medium	Commonly used as a reagent in metallurgical processes [57]	Environmentally very harmful Requires constant integration of the input into the circuit More complex pH control [56]

**Table 4 nanomaterials-14-00069-t004:** Leaching agents used in the hydrometallurgical process.

Leaching Agent	Metal	Concentration of Metal from Leaching (mg/L or Leaching Percentage)	Reference
Aqua regia	Au	125	[50]
	Pd	13	
	Ag	163	
Aqua regia	Au	57%	[51]
	Cu	44%	
Aqua regia	Au	550	[52]
Thiourea	Au	100%	[53]
	Ag	81%	
	Pd	13%	
Halogens (Iodine)	Au	99.8%	[53]
	Ag	81.7%	
	Pd	74%	
Iodine-iodide	Au	98%	[54]
KI	Au	99.2%	[55]
Thiosulfate	Au	81%	[58]
	Ag	88%	
	Cu	32%	

**Table 5 nanomaterials-14-00069-t005:** Green recovery process and microorganisms used for metal recovery from e-waste.

Green Recovery Method	Microorganism	Metal	Concentration of Metal in Leaching (% or mg/L)	Operation	Reference
Biohydrometallurgy	*Acidithiobacillus ferrooxidans* and *Acidiphilium acidophilus*	Cu	96%	Concentrated by fractional chemical precipitation	[19]
		Zn	94.5%		
		Ni	75%		
		Pb	74.5%		
Bioleaching	*Acidithiobacillus ferrooxidans*, *Leptospirillum ferrooxidans*, and *Acidithiobacillus thiooxidans*	Cu	95%	Concentrated by electrowinning	[29]
Bioleaching	*Leptospirillum ferriphilum*	Fe	8000 mg/L	pH 1.2 T: 35 °C	[79]
Bioleaching	*Pseudomonas fluorescens*	Au	54%	—	[69]
Bioleaching	*Chromobacterium violaceum*	Au	11–30%	—	[62]
Hybrid combination: Bioleaching-Biosorption	*Lactobacillus acidophilus*	Au	85%	5 mL of microbial inoculum and 1 g of contact material 90 days	[80]
Bioleaching	*Aspergillus niger*	Cu	100%	pH 7.0, 5 days, T: 30 °C, 200 rpm Pulp density: 0.5%	[81]
Bioleaching	*Aspergillus niger*	Ag	67%	—	[67]
		Cu	50%		
Bioleaching	*Aspergillus niger* consortium	Au	87%	—	[70]
		Cu	81.7%		
		Ni	74%		
Hybrid combination: Bioleaching-Biosorption	*Pleurotus florida* and *Pseudomonas* spp.	Cu	18%	1 g biomass pH 7.2 T: 27 °C Lacase production	[75]
		Fe	12.4%		
Biosorption	*Aspergillus carbonarius*	Cr^6+^	92.43%	pH 2.0 during 12 h at 37 °C	[74]
Bioaccumulation	*Penicillium expansum*	La	390 mg/L	—	[82]
		Tb	1520 mg/L		
Biosorption	*Aspergillus* spp.	Cd	88%	pH 4 T: 30 °C	[73]

**Table 6 nanomaterials-14-00069-t006:** Concentration and purification of metals from WPCBs for nanoparticle production.

Method	Conditions	Element	Reference
Fractional chemical precipitation, solvent extraction, and electrowinning	A two-stage process for the separation of Cu using a phenolic oxime dissolved in kerosene as an organic liquid phase for the recovery of copper and nickel from the leach liquor. The availed residue is dissolved in the halide salts in the presence of acidic conditions, showing the quantitative dissolution of gold and silver. Solvent extraction with amide-based reagents recovered gold by leaving the silver-rich raffinate. A precipitation or cementation technique using copper powder is implemented for the precipitation of silver.	Cu, Ni, Ag, Au	[19]
Solvent extraction (SX)	Solvents: organophosphorus derivatives, guanidine derivations, and a mixture of amines–organophosphorus derivatives prior to chemical reduction.	Au	[87]
Electrowinning (EW)	Solutions: 8.1 g/L Cu and 9.1 g/L Fe Cathode: copper Anode: Ti-metal oxide Current density of 300 A/m^2^	Cu (66%)	[29]
Electrowinning (EW)	Cathode and anode: Au, 100 AC voltage for 5 Solution: 4-(2-hydroxyethyl)-1-piperazineethanesulfonic acid (HEPES) buffer solution containing 1,2-dioleoyl-sn-glycero-3-phosphocholine (DOPC)	Au	[88]
Dialysis	Polyethyleneimine (PEI) membrane	Nd Pr	[85]

**Table 7 nanomaterials-14-00069-t007:** Studies for nanoparticle reduction employing different algae.

Algae	Highlights	Element	Nanoparticle Size	Reference
*Sargassum* spp.	The microwave-assisted synthe-sis (MAS) of AgNPs using Sar-gassum spp. biomass involves the reduction of Ag+ ions by diverse organic compounds present in the macroalgae extract, including polysaccharides, proteins, poly-phenols, and flavonoids, through redox reactions.	Ag	10 to 175 nm. Average size of 36.43 nm	[115]
*Rhizoclonium hieroglyphicum*, *Lyngbya majuscule* and *Spirulina subsalsa*	The reduction of gold particles is attributed to the presence of cellular reductases, with proteins such as cysteine serving to stabilize the nanoparticles.	Au	<20 nm	[116]
*Chlorella vulgaris*	Carboxylate groups on the surface of *Chlorella vulgaris* cells capture metal ions, which are subsequently reduced to silver nanoparticles by reductase enzymes.	Ag	Average size 10.95 nm	[117]

## Data Availability

Not applicable.

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
