# Peer review of "From E-Waste to High-Value Materials: Sustainable Synthesis of Metal, Metal Oxide, and MOF Nanoparticles from Waste Printed Circuit Boards"

_nanomaterials, 2023, doi:10.3390/nano14010069_

Round 1

Reviewer 1 Report

Comments and Suggestions for Authors

This paper explores the pressing issue of electronic waste (E-waste) and, in particular, the sustainable synthesis of metal nanoparticles from Waste Printed Circuit Boards (WPCBs). The topic is both timely and critical given the exponential growth of E-waste and the valuable resources contained in WPCBs. Here are some suggestions and feedback:

Clarity and Organization: The introduction effectively sets the stage for the discussion. However, the text could benefit from more precise subheadings to guide the reader through the paper. Consider breaking down the content into sections, such as "Metal Recovery from WPCBs," "Green Recovery Systems," and "Applications of Nanoparticles," for clearer navigation.

Citations and References: Ensure that all claims and statements are properly cited and referenced. Readers should be able to trace the sources of information for a comprehensive understanding of the topic. some important literature should be cited. such as Trends in Chemistry, vol. 4, no. 12, pp. 1065 – 1077 (2022). Science China Materials, Vol 65, pp. 2445–2452 (2022)

Future Work: Consider dedicating a small section to future research directions in this field. What are the emerging trends and areas where further investigation is needed?

Overall, this paper addresses an important and timely issue. With some organization, additional data, and a few adjustments to the presentation, it can make a significant contribution to the discussion on E-waste recycling and sustainable nanotechnology.

Comments on the Quality of English Language

no necessary

Author Response

Thank you to the reviewer for your thoughtful and constructive feedback on the manuscript. Your insights have greatly contributed to its improvement. Below, you will find point-by-point responses addressing each of your comments.

Clarity and Organization: The introduction effectively sets the stage for the discussion. However, the text could benefit from more precise subheadings to guide the reader through the paper. Consider breaking down the content into sections, such as "Metal Recovery from WPCBs," "Green Recovery Systems," and "Applications of Nanoparticles," for clearer navigation.

R:/ The adjustments to the sections within the introduction were applied as suggested by the reviewer.

Citations and References: Ensure that all claims and statements are properly cited and referenced. Readers should be able to trace the sources of information for a comprehensive understanding of the topic. some important literature should be cited. such as Trends in Chemistry, vol. 4, no. 12, pp. 1065 – 1077 (2022). Science China Materials, Vol 65, pp. 2445–2452 (2022).

R:/ The authors ensured that all claims and sentences were correctly cited and referenced in the article. Additionally, the sources suggested by the reviewer were analyzed: the publication Trends in Chemistry, entitled "Recommended practices and benchmarking of foam electrodes in water splitting", is an opinion article that we do not consider relevant to the focus of the review, which deals with the sustainable production of nanoparticles from WPCBs. On the other hand, the Science China Materials publication is about Direct observation of dynamic surface reconstruction and active phases on honeycomb Ni3N−Co3N/CC for oxygen evolution reaction, which is not part of the focus of this review.

Future Work: Consider dedicating a small section to future research directions in this field. What are the emerging trends and areas where further investigation is needed? Overall, this paper addresses an important and timely issue. With some organization, additional data, and a few adjustments to the presentation, it can make a significant contribution to the discussion on E-waste recycling and sustainable nanotechnology.

R:/ A paragraph was added to the conclusions indicating considerations and future work derived from this review, as suggested by the reviewer.

Reviewer 2 Report

Comments and Suggestions for Authors

The subject of the review is topical and is worth reviewing.  The title needs modifying as the text discusses not just metal nanoparticles, but also metal oxide and MOF nanoparticles. Thus it should refer to metal and metal-containing nanoparticles.  However the account provided is deficient in a number of ways and must be improved considerably to be acceptable. It is not a critical review and it appears to be quoting opinions often from other reviews without critically examining  these views. There are also contradictions within paragraphs wherein a generalisation is then contradicted by later material. The detailed consequences of "green" procedures are not considered. For example when promoting use of plant extracts, the amount of active reagent available from the plant as a percentage by weight, the cost of extraction, the solvent needed and energy involved  and the amount of waste generated are not considered. It may be viewed as green waste but it still has to be dealt with. Bacteria and fungi may appear attractive and "green", but what are the costs in obtaining them, what is their efficiency, how are metals recovered from them, and can the bacteria and fungi be recycled? These issues need to be recognised and either  be addressed or raised as issues needing resolution. Two reviews on recyling of e-waste need to be included  .      Grimes, S. M.; Maguire, D. Assessment of priorities in critical material recovery from Waste Electrical and Electronic Equipment. Resource. Policy, 2020, 68, 101658.      Mir, S.; Dhawan, N. A comprehensive review on the recycling of discarded printed circuit boards for resource recovery. Resources, Conserv. Recycl. 2022, 178, 106027. The first of these considers economic issues.

On p1 WPCBs are about 10% of electronic waste but on p3, they are 3-5%. In para 1 p3, Cu is a primary metal (base metal?), but in the section under Table 1, it appears to have become a precious metal, as if pins are covered by Ni, we have Cu,Sn, Zn, Ni only as possible precious metals. % compositions are presented in two cases in ascending order and then we have 0.1 to less than 0.1%,. Why not < or =0.1%.. Again under Table 1 "are used for conducting electricity and fixing."  Fixing what? Line 115 "disassembly" and "disassembling"  both used in the one sentence, yet both have the same meaning. Line 117 "at the beginning" should be a new sentence.  line 120/1 ".where polymeric components are separated into metallic components" is a nonsense  Lines 128/9 are repeated in lines 132-4

Lines 171-3, A list of leaching agents are given, of quite different chemical properties and  hazards, but there is no indication of their respective merits and demerits, let alone their uses. 

Heading 4 "Recovering system" should be "Recovery Systems" The long sentence line 166-190  needs rewording as it does not make sense.  The next sentence lines 190/1 about metals being recovered from solution by "microbial oxidation" makes no chemical sense. On extraction metals are usually in their highest stable oxidation state, how can they be recovered (as M ?) by oxidation?

Lines 197/8 The terms "redoxolysis" and "complexolysis" should by sent to non-recyclable waste. New words have value if they serve a useful purpose where no alternatives are available.  In this case "redox" and "complexation"  are well established and correct and should be used.

4.3.2  Biosorption  is said to be a passive process (line 243), but in lines 265-270. there is a description of functional groups in the biological materials that can coordinate  to metals i. Their role is not biosorption but chemical. Lines 253-256 describe capacity to absorb Cr(VI). What then happens to the contaminated biomass produced? Lines 272/3 "At lower cell densities , biomass absorbs more metal ions owing to electrostatic interactions between cells."  What does this mean? At lower cell densities, electrostatic interactions between cells should be less.  If metal ions are being absorbed owing to an electrostatic process, this is hardly a passive process, but an ionic interaction (which is one extreme coordination model). Lines 281-283 and Table 4, what happens next to the biomass containing metal ions, and, if they are then leached off . is the biomass recycled or is there a disposal problem at this point? Column heading on Table 4 "metal on" Should this be "meta; ion"? Line 292 "no" should be "non"? Lines 295 and 299, solvent and extractant are used interchangeably. but are not necessarily the same, as the extractant is often dissolved in an inert solvent such as kerosene. Lines 315/316. Which reference specifically deals with use of dialysis to concentrate rare earth ions?  Lines 334-348, same question as before, how are the metal ions recovered from the biomass, and is it recycled or what about disposal?  line 346 "substrates" not singular. line 359 define PVP.

What is the cost of ascorbic acid and is it affordable to be used as an industrial scale reductant? How does it inhibit nanoparticles interacting with the environment? 

 6.2 Plants  How much reductant is available in terms of % mass, how effective is the extracted reductant compared to commercial chemical use, what happens to the plant residues, and what % of the original plant material becomes waste.? Lines 379-381 Since CuO nanoparticles have Cu in the maximum usual oxidation state, how is biogenic reduction by the plant leaf extract involved? Lines 385-9 Similar issue CuO NPs from Cu(NO3)2 is not a reduction process so what is the role of the plant extract? Lines 393-6 This sentence does not make sense. Lines 413-415. How can Ti and TiO2 be prepared usefully from PCBs when only present in trace amounts (Table 1) and are readily available and relatively inexpensive. Lines 423-4 "encapsulating nanoparticles within  vesicles for later release. How is it known that the metals are present as nanoparticles and how are they released as nanoparticles? Line 429 Is methylation of Au detected or is this just an hypothesis? Methylation would enhance gold toxicity. Lines 438-9 "bacterial fermentation processes often involve multiple steps......a clear colloidal broth." Are multiple steps a disadvantage in using fermentation, what is in the colloidal broth to make it relevant? Fig. 6 on Biosynthesis of nanoparticles. Reduction is obviously relevant but what about chelation. As listed on the Fig., it follows reduction?

Lines 471-2 appear to be contradicted by three examples given in line 462-479.

Table 4  The carbonyl group listed as a reductant should be removed as it can only act in this way under extreme conditions. 

Industrial applications: It should be made clear at the outset that these applications are not specific to metals extracted from e-waste, but are possible with any source of the metals, but in the examples, the researchers have used e-waste derived material.

Lines 531-8 Is a TiO2 induced example relevant. Ti/TiO2 are a very minor component of WPCBs (Table 1) and its separation in a significant amount would be tedious and expensive, Lines 538-541  Synthesis of nanoparticles does not have a considerable effect on photocatalysis, nanoparticles do. Lines 552-3 Insert "a" before "Cu/CuO blend" Lines 556-562  This needs qualifying as Na2edta will remove all base metals not just Sn. Lines 563-572 Is this really a good example. Cu from WPCBs is only one component of several chemicals , e.g. 2-methylimidazole and titanium tetraethoxide, needed to make the Cu spiked MOF. In addition, the experimental  account of the synthesis of the MOF contains an experimental nonsense relating to the Cu from WPCBs. "021g Cu (recovered from WPCBs) was dissolved in 3 mL of pure water." Cu metal does not dissolve in water.. Line 580 "Ferrous nanoparticles."  What does this mean? Lines 595-6  Se nanoparticles from e-waste. Table 1 does not list Se in e-waste

Conclusions 1st sentence  Elements recovered from e-waste can be used for nanoparticle synthesis for industrial applications. In most cases cited this was not the purpose of the study. Lines 612/3 These conclusions need qualifying as outlined earlier. Line 616 "Solvent extraction with specific solvents....." See comments above. It is normally extraction with a specific reagent in an inert solvent not extraction with a neat extractant.

The references need considerable attention and have not been properly checked.  Thus refs 12, 29, 33, 37,41, 45, 46, 56,87, 89, 90, 93, 95, 97,  are incomplete.

The review might be simplified by restricting the discussion to recovery of Cu, Ag, Au from e-waste, as these are the more valuable components  However another point also needs addressing, once these elements even all those in Table 1 have been removed, what happens to the residual e-waste, as it is the largest amount by weight of the WPCBs?

Comments on the Quality of English Language

The English is mostly OK, and errors noted may arise from incomplete proof reading rather than inherent deficiencies. The grammar is overall excellent.

Author Response

Thank you to the reviewer for your valuable feedback on the manuscript. Below, you will find point-by-point responses addressing each of your comments. Your input has been instrumental in enhancing the quality of the manuscript.

The subject of the review is topical and is worth reviewing.  The title needs modifying as the text discusses not just metal nanoparticles, but also metal oxide and MOF nanoparticles. Thus it should refer to metal and metal-containing nanoparticles.

R:/ The title of the manuscript was modified according to the recommendations proposed by the reviewer. A mention of metal oxide nanoparticles and MOFs was included. The suggested title retains the same essence, under this proposal: “Transforming E-Waste into High-Value Materials: Sustainable Synthesis of Metal, Metal Oxide and MOF Nanoparticles from Waste Printed Circuit Boards

The detailed consequences of "green" procedures are not considered. For example when promoting use of plant extracts, the amount of active reagent available from the plant as a percentage by weight, the cost of extraction, the solvent needed and energy involved  and the amount of waste generated are not considered. It may be viewed as green waste but it still has to be dealt with. Bacteria and fungi may appear attractive and "green", but what are the costs in obtaining them, what is their efficiency, how are metals recovered from them, and can the bacteria and fungi be recycled? These issues need to be recognised and either  be addressed or raised as issues needing resolution.

R:/ We appreciate your feedback and the opportunity to address the raised concerns regarding the detailed implications of "green" procedures in the recovery of metals from electronic waste.

We understand the importance of considering critical aspects such as the weight percentage of the active principle available in plants, extraction costs, the type of solvent used, the energy involved, and the amount of generated waste. We acknowledge that these are key factors in assessing the sustainability of the proposed methods. However, green chemistry, at its core, focuses on industrial applications, and the approach to chemical synthesis may vary depending on the type of compound of interest and the associated plant source. These aspects introduce variability in the mentioned metrics, making the generalization of specific values challenging.

In general terms, we can emphasize that green processes, such as bioleaching, tend to require relatively low capital investment, as biological and microbiological sources can be isolated from process wastewater. Additionally, these processes exhibit reduced operating costs compared to conventional approaches, contributing to lower environmental pollution during the process. It is crucial to note that green chemistry aims to minimize risks and maximize efficiency, and in this regard, bioleaching presents notable advantages in terms of sustainability and low environmental impact.

Concerning the bacteria and fungi used in these processes, we understand the importance of evaluating their efficiency and the possibility of recycling. Although these aspects may vary depending on the specific application and process conditions, we recognize the need to address these issues and provide a detailed assessment in future research.

Two reviews on recyling of e-waste need to be included. Grimes, S. M.; Maguire, D. Assessment of priorities in critical material recovery from Waste Electrical and Electronic Equipment. Resource. Policy, 2020, 68, 101658.      Mir, S.; Dhawan, N. A comprehensive review on the recycling of discarded printed circuit boards for resource recoveryResources, Conserv. Recycl. 2022, 178, 106027. The first of these considers economic issues.

R:/ The authors have reviewed these two articles suggested by the reviewer, and have found them relevant for inclusion in the review. The central approaches of both were revised and included in the final version.

On p1 WPCBs are about 10% of electronic waste but on p3, they are 3-5%. In para 1 p3, Cu is a primary metal (base metal?), but in the section under Table 1, it appears to have become a precious metal, as if pins are covered by Ni, we have Cu,Sn, Zn, Ni only as possible precious metals. % compositions are presented in two cases in ascending order and then we have 0.1 to less than 0.1%,. Why not < or =0.1%..

R:/ The information was corroborated and some modifications were made to the text, the word precious metal was changed to valuable metals.

Again under Table 1 "are used for conducting electricity and fixing."  Fixing what? Line 115 "disassembly" and "disassembling"  both used in the one sentence, yet both have the same meaning. Line 117 "at the beginning" should be a new sentence. 

R:/ It was already deleted to make sense.

line 120/1 ".where polymeric components are separated into metallic components" is a nonsense  Lines 128/9 are repeated in lines 132-4

R:/ The word into was changed to “from”

Lines 171-3, A list of leaching agents are given, of quite different chemical properties and  hazards, but there is no indication of their respective merits and demerits, let alone their uses. 

R:/ A new Table with the list of leaching agents indicating merits and demerits was added to the manuscript.

Heading 4 "Recovering system" should be "Recovery Systems" The long sentence line 166-190  needs rewording as it does not make sense. 

R:/ The term "Recovering system" was changed to "Recovery Systems" in the final version of the manuscript, following the recommendations proposed by the reviewer. Thank you very much.

The next sentence lines 190/1 about metals being recovered from solution by "microbial oxidation" makes no chemical sense. On extraction metals are usually in their highest stable oxidation state, how can they be recovered (as M ?) by oxidation?

R:/ The sentence contained a typographical error. The correct term is microbial reduction

Lines 197/8 The terms "redoxolysis" and "complexolysis" should by sent to non-recyclable waste. New words have value if they serve a useful purpose where no alternatives are available.  In this case "redox" and "complexation"  are well established and correct and should be used.

R:/ The terms "redoxolysis" and "complexolysis" were changed to "redox" and "complexation" throughout the document, following the recommendations received by the reviewer.

4.3.2  Biosorption  is said to be a passive process (line 243), but in lines 265-270. there is a description of functional groups in the biological materials that can coordinate  to metals i. Their role is not biosorption but chemical.

R:/ Dear reviewer, the cell wall of microorganisms like yeast or microalgae contains different polysaccharides, which have ion exchange properties.  The divalent ions (cations) of the cell wall are replaced by the ions (protons and/or cations) of heavy metals, or by the formation of a complex between the metal ions and the functional groups of the cell wall.

Lines 253-256 describe capacity to absorb Cr(VI). What then happens to the contaminated biomass produced?

R:/ The biomass can in principle be regenerated by a desorption process using weak acids. This allows it to be reactivated for reuse and, at the same time, to obtain a metal-rich stream that can be recovered by chemical or electrochemical reduction methods.

Lines 272/3 "At lower cell densities , biomass absorbs more metal ions owing to electrostatic interactions between cells."  What does this mean? At lower cell densities, electrostatic interactions between cells should be less.  If metal ions are being absorbed owing to an electrostatic process, this is hardly a passive process, but an ionic interaction (which is one extreme coordination model).

R:/ We appreciate the evaluator's detailed review in this regard. We found a typographical error since in the source of information this phenomenon is explained considering electrostatic interactions that are strengthened at low pH levels. This is why we proceeded to correct it in the manuscript.

 Lines 281-283 and Table 4, what happens next to the biomass containing metal ions, and, if they are then leached off . is the biomass recycled or is there a disposal problem at this point? Column heading on Table 4 "metal on" Should this be "meta; ion"? Line 292 "no" should be "non"?

R:/ It was changed and your suggestion was considered. The majority of waste generated in green metal extraction processes, as well as what is derived from the synthesis of NPs, can be treated or valorized through three major processes: thermochemical, biological and mechanical. The first include pyrolysis, gasification, combustion, among others, the biological processes include fermentation, digestion and production of enzymes and finally the mechanical ones refer to operations, drying, grinding and pelletizing, where added value can be given. to said waste. However, thermochemical processes are the most required, due to the reduction in waste volumes and the recovery of molten metals.

Lines 315/316. Which reference specifically deals with use of dialysis to concentrate rare earth ions? 

R:/ The reference which specifically deals with use of dialysis to concentrate rare earth ions is the following: Hammache, Z.; Bensaadi, S.; Berbar, Y.; Audebrand, N.; Szymczyk, A.; Amara, M. Recovery of Rare Earth Elements from Electronic Waste by Diffusion Dialysis. Sep Purif Technol 2021, 254, 117641, doi:https://doi.org/10.1016/j.seppur.2020.117641.

Lines 334-348, same question as before, how are the metal ions recovered from the biomass, and is it recycled or what about disposal?  line 346 "substrates" not singular. line 359 define PVP.

R:/ Thank you for the suggestion. The definition was added, PVP is the Polyvinylpyrrolidone polymer.

What is the cost of ascorbic acid and is it affordable to be used as an industrial scale reductant? How does it inhibit nanoparticles interacting with the environment? 

R:/ The cost of ascorbic acid is ranging between US$ 2.4 a 4.0 per kilogram, which is less than the cost of US$ 25 a 30 per kilogram of sodium borohydride which is the industrial raw material used currently. Ascorbic acid in nanoparticle reduction does not interfere with the applications but rather improves certain aspects. Elshoky 2018 found that ascorbic acid prevented cellular uptake and improved the biocompatibility of chitosan nanoparticles (DOI: 10.1016/j.ijbiomac.2018.04.055).

 6.2 Plants  How much reductant is available in terms of % mass, how effective is the extracted reductant compared to commercial chemical use, what happens to the plant residues, and what % of the original plant material becomes waste.?

R:/ A paragraph has already been added talking about this topic.

Lines 379-381 Since CuO nanoparticles have Cu in the maximum usual oxidation state, how is biogenic reduction by the plant leaf extract involved? Lines 385-9 Similar issue CuO NPs from Cu(NO3)2 is not a reduction process so what is the role of the plant extract?

R:/ The cited reference indicates that the plant extract performs both the role of reducing agent and stabilizing agent from the selectively leached solution of WPCB to produce nanoparticles, The amide groups present in proteins and enzymes of leaf extract are responsible for the reduction process and amine groups of leaf extract which acts as a capping agent.

Additionally, it gives an approach to how the waste generated from the nanoparticle synthesis process is treated, through a thermal process to produce grounded char resulting in the formation of graphene oxide (GO).

In the case of CuO NPs from Cu(NO3)2, the hydroxyl groups that are present in all phenolic acids, reduce the Cu ions, Therefore polyphenol acts as a reducing agent.

 Lines 393-6 This sentence does not make sense.

R:/ This phrase was reviewed and therefore deleted in the final version.

Lines 413-415. How can Ti and TiO2 be prepared usefully from PCBs when only present in trace amounts (Table 1) and are readily available and relatively inexpensive.

R:/ Although the WPCB has traces of Ti and TiO2, these, due to their chemical characteristics, could present some degree of photocatalytic effect. However, the focus of this review is to highlight how metallic nanoparticles enhance the effects of materials such as TiO2 photocatalysis.

Lines 423-4 "encapsulating nanoparticles within vesicles for later release. How is it known that the metals are present as nanoparticles and how are they released as nanoparticles?

R:/ Dear reviewer, extracellular vesicles are produced by different species. In Gram-negative bacteria, membrane vesicles (MVs) can originate either from blebs of the outer membrane or from endolysin-triggered explosive cell lysis, which is often induced by genotoxic stress as a presence of metals. Although less is known about the mechanisms of vesiculation in Gram-positive and Gram-neutral bacteria, recent research has shown that both lysis and blebbing mechanisms also exist in these organisms. The presence of nanoparticles inside the vesicles can be observed by microscopy (SEM or TEM) and their subsequent release can occur using acids or physical mechanisms such as ultrasonic methods in high frequencies.

Line 429 Is methylation of Au detected or is this just an hypothesis? Methylation would enhance gold toxicity.

R:/ According to the referenced article, they do not verify the methylation but establish it as a possibility supported by another observation in a similar study done by Reith (2009) (Source: https://doi.org/10.1073/pnas.0904583106).

Lines 438-9 "bacterial fermentation processes often involve multiple steps......a clear colloidal broth." Are multiple steps a disadvantage in using fermentation, what is in the colloidal broth to make it relevant?

R:/ In relation to cultures using fungi, the separation of the biomass from the broth is much simpler, and fewer unit operations are required, which makes the process more agile. In general, fungi are filamentous by nature and able to withstand the pressure of flow and mixing in the bioreactor trough. In bacterial culture, obtaining a broth separated from biomass involves numerous additional steps, including filtration, solvent extraction and the use of sophisticated apparatus that increased considerable investment costs for equipment.

Fig. 6 on Biosynthesis of nanoparticles. Reduction is obviously relevant but what about chelation. As listed on the Fig., it follows reduction?

R:/ When the nanoparticle is synthesized by the chelation mechanism, the metal atom is linked by coordinated bonds to organic species, which increases its thermodynamic stability.

Lines 471-2 appear to be contradicted by three examples given in line 462-479.

R:/ The mechanism used by fungi for the reduction of metals and synthesis of nanoparticles is ideal and widely investigated, but specifically, few works have been carried out using WPCBS as a substrate. The idea is to invite researchers to join in using this pathway and take advantage of the extracellular synthesis offered by various fungi and the resistance to inhibition presented by multiple metals on them.

Table 4 The carbonyl group listed as a reductant should be removed as it can only act in this way under extreme conditions.

R:/ The carbonyl group was removed.

Lines 556-562  This needs qualifying as Na2edta will remove all base metals not just Sn.

R:/ According to the results from (Abdo, D.M.; Abdelbasir, S.M.; El-Sheltawy, S.T.; Ibrahim, I.A. Recovery of Tin as Tin Oxide Nanoparticles from Waste Printed Circuit Boards for Photocatalytic Dye Degradation. Korean Journal of Chemical Engineering 2021, 38, 1934–1945, doi:10.1007/s11814-021-0838-9), the recycle of WPCBs for their metallic content and recovery from this toxic material value-added tin oxide (SnO2) nanoparticles was achieved via selective leaching in Na2-EDTA chelating agent and compare with usual acid leaching.

Cu metal does not dissolve in water.

R:/ Thank you for the comment. What the article means is that it is better dispersed.

Line 580 "Ferrous nanoparticles."  What does this mean?

R:/ Dear reviewer, it was a translating mistake. The correct term is iron nanoparticles.

Lines 595-6  Se nanoparticles from e-waste. Table 1 does not list Se in e-waste

R:/ Selenium may be present in small concentrations in the typical composition of scrap metal. For example, in the study cited above it is present and is used for biomedical applications.

The references need considerable attention and have not been properly checked.  Thus refs 12, 29, 33, 37,41, 45, 46, 56,87, 89, 90, 93, 95, 97,  are incomplete.

R:/ The cited references were properly checked and adjusted in order to complete the missing information. We believe that the reference manager (Mendeley) failed to update it, so we decided to rewrite it manually.

The review might be simplified by restricting the discussion to recovery of Cu, Ag, Au from e-waste, as these are the more valuable components 

R:/ We appreciate the reviewer suggestion on this particular aspect. However, we strongly believe that restricting the discussion to recovery of Cu, Ag, Au from e-waste excludes some important findings about other relevant nanoparticles such as nickel or selenium, among others.   

However another point also needs addressing, once these elements even all those in Table 1 have been removed, what happens to the residual e-waste, as it is the largest amount by weight of the WPCBs?

R:/ The residual e-waste, representing the largest amount by weight of the WPCBs, undergoes various beneficial processes. Notably, the non-metallic fraction of PCB waste (NMF) plays a key role in this scenario. After the extraction of metals through leaching processes, the remaining e-waste, rich in non-metallic components, is repurposed for different applications such as in phenolic moulding compounds (PMC), construction materials like wood-plastic composites, and even in the production of porous carbons with high surface areas for adsorption applications, as suggested by Hadi et al (2015) [Reference: https://doi.org/10.1016/j.jhazmat.2014.09.032].

Reviewer 3 Report

Comments and Suggestions for Authors

The article provides a comprehensive review of pathways for synthesizing metal nanomaterials from WPCBs with an emphasis on green bio-based methods. However, it may lack an essential analysis of the characterization of these resulting nano-materials and associated performance in their applications. Besides, it also needs to provide an objective and balanced evaluation regarding both the advantages and disadvantages of the strategy. The following are for revisions.

In line 14 and line 45, the high content of valuable metals and REEs is misleading, it could be true when compared with mineral ores. REEs are normally at low concentrations except for some special applications.

In line 87, the primary focus on WPCBs from PCs is not evident in the article.

In line 96, The illustration of metal recovery in Figure 1 is not proper, instead, according to the review, it should be more chemical rather than mechanical.

In line 103 and table 1, the classification of elements by the percent is not correct.

In line 119, the advanced sorting was overlooked, such advances can be found in https://doi.org/10.1016/j.cej.2022.135886, https://doi.org/10.1016/j.resconrec.2023.107033, etc.

In line 152, can you justify why only froth flotation is discussed at the beginning of the section? I think it should be categorized into the metal enrichment section. Besides, Figure 3 doesn't show the explicit process of the froth flotation.

Rephrase from line 186 to line 190.

In line 201, Figure 4 doesn't provide an in-depth demonstration of bioleaching mechanisms.

In line 534, the citation format is not proper.

In section 7.4, is there any concern about the impurity content from WPCBs for biomedical applications?

In line 601, it was explored...

Comments on the Quality of English Language

Minor revision.

Author Response

Thank you to the reviewer for your valuable feedback on the manuscript. Below, you will find point-by-point responses addressing each of your comments. Your input has been instrumental in enhancing the quality of the manuscript.

In line 14 and line 45, the high content of valuable metals and REEs is misleading, it could be true when compared with mineral ores. REEs are normally at low concentrations except for some special applications.

R:/ To avoid the phrase related to REEs being interpreted as misleading, it was specified that their high concentration is relative to the concentration of metallic ores.

In line 87, the primary focus on WPCBs from PCs is not evident in the article.

R:/ Dear reviewer, throughout the manuscript we focus on the composition of WPCBs, the methods of extracting valuable metals from them and the possibilities for the production of metal nanoparticles. We consider that WPCBs had an important representation throughout the manuscript, since we did not focus on other typical waste types within the E-waste family.

In line 96, The illustration of metal recovery in Figure 1 is not proper, instead, according to the review, it should be more chemical rather than mechanical.

R:/ Many thanks to the reviewer for the recommendation received. As shown in the sections of the review, both physical and chemical methods are shown in detail. In fact, the chemical methods are developed in greater detail. The figure is a descriptive synthesis.

In line 103 and table 1, the classification of elements by the percent is not correct.

R:/ Thanks for the observation, the text and table were corrected in this new version for the manuscript.

In line 119, the advanced sorting was overlooked, such advances can be found in https://doi.org/10.1016/j.cej.2022.135886, https://doi.org/10.1016/j.resconrec.2023.107033, etc.

R:/ We added a paragraph remarking recent advances in E-waste sorting, as follows: Innovative approaches that integrates optical sorting, X-ray spectroscopy, and various physico-mechanical methods have been implemented to achieve effective metal pre-concentration for recycling. The optical sorting of ceramic capacitors (ECs) is detailed, employing a conveyor belt prototype equipped with machine vision techniques, including a convolutional neural network (CNN) written in Python. Electro-pneumatic nozzles then guide recognized ECs into sorting bins based on directional airflow, with recovery rate and sorting accuracy serving as evaluation metrics. Additionally, the authors describe the application of ME-XRT (multi-energy X-ray transmission) sorting for super-large ceramic capacitors (SLCCs), outlining the interpretation of element K-edges from the XRT spectrum and the use of a Python code to simulate sorting performance. The article concludes with the ongoing implementation of ME-XRT sorting onto a conveyor system, pending X-ray device operation licensing [10.1016/j.cej.2022.135886].

In line 152, can you justify why only froth flotation is discussed at the beginning of the section? I think it should be categorized into the metal enrichment section. Besides, Figure 3 doesn't show the explicit process of the froth flotation.

R:/ Dear reviewer, froth flotation emerges as a pivotal and indispensable stage in the metal beneficiation processes from WPCBs. Its significance lies in the efficient separation of plastic residues from metallic fractions. Leveraging the high hydrophobicity of nonmetallic particles, froth flotation facilitates the attachment of these particles to bubbles, causing them to float. In contrast, metal particles, characterized by reduced hydrophobicity, sink during the process. This inherent property ensures a precise and effective separation between plastic and metal components within WPCBs. The utilization of froth flotation not only enhances the purity of the recovered metal fractions but also streamlines downstream processes, contributing to the overall efficiency of WPCB recycling strategies.

Rephrase from line 186 to line 190.

R:/ The text was rephrased as follows: “Biohydrometallurgy, an interdisciplinary field merging biology with the hydrometallurgical process, employs selective and environmentally friendly approaches for metal recovery [35]. Utilizing agents like fungi, bacteria [15], algae, or fermentation products such as enzymes, it transforms specific substrates into soluble salts within an aqueous medium. Microbial reduction facilitates the recovery of soluble heavy metals [36]. Subsequently, the key methods in this process are highlighted.”

In line 201, Figure 4 doesn't provide an in-depth demonstration of bioleaching mechanisms.

R:/ The figure was improved to provide an in-depth demonstration of bioleaching mechanisms.

In line 534, the citation format is not proper.

R:/ The citation was improved.

In section 7.4, is there any concern about the impurity content from WPCBs for biomedical applications?

R:/ Dear reviewer, to clarify issues related to the impurity content of WPBCs for biomedical applications, we have included the following text in the review at the end of section 7.4:

"Nanoparticles must be non-toxic and biocompatible to be successful in biomedical applications. Nanoparticles can easily enter the body due to their small size and reach very sensitive organs by different pathways. The cytotoxicity of nanoparticles depends on several parameters such as size, shape, surface charge, chemistry and surface modifications [1,5L].

Nanoparticles have found wide biomedical application due to their physicochemical and behavioral uniqueness, although concerns about their toxic effects on the biological system are now attracting the attention of the global health community. This calls for the importance of studying and understanding the effects based on the cellular and molecular mechanisms by which they cause these effects. Some toxic mechanisms identified are through induction of free radicals (ROS), cytotoxicity to cells and genotoxic and neurotoxic effects. This toxic effect depends on the type of nanoparticles, size, surface area, shape, aspect ratio, surface coating, crystallinity, dissolution and agglomeration [1,4L].

Nanotoxicological studies are planned to determine to what extent these properties could constitute an attack on the atmosphere, animals and humans. For example, nanoparticles are known to cause damage to the central nervous system, circulatory system, respiratory system and cardiovascular system [1,2L], although this will depend largely on the type of application and how they are synthesized."

In line 601, it was explored...

R:/ The text was rewritten to improve its comprehension.

Reviewer 4 Report

Comments and Suggestions for Authors

In this manuscript, Pineda-Vásquez et al. has presented the review on the recycling of waste printed circuit boards (WPCBs) specifically using the biohydrometallurgy/bioleaching and biosorption method. Moreover, authors have discussed on the synthesis of nanoparticles from the WPCBs employing various green methods. Overall, this review topic is interesting and may add knowledge to the literature on the recovery of critical elements from WPCBs. However, the reviewer has provided several comments for enhancement of the overall quality of the review prior consideration for possible publication.

Comments:

1.       Abstract: At the end of the abstract, add a statement on the important implication of this review.

2.       Line 46: “rare earth elements”. Could you list the dominant elements here.

3.       Line 52 – 56: Need reference for the statements.

4.       In introduction, authors should strengthen their argument why it is necessary to recycle WPCBs from the perspectives of environmental benefits, resource recovery and circularity.

5.       In the last paragraph of the introduction, justification is sought on the novelty and importance of this review with respect to the existing published review articles. Why this review should be published and how it will contribute to the academics and industries who are working very hard for developments of novel WPCBs recycling methods for optimum recovery of valuable metals.

6.       In the introduction or as a separate section, develop a section on the “Review Method” by explaining the selection criteria such as the scientific database (Scopus, Web of Science and/or Google Scholar, etc.), the relevant keywords, the publication period (e.g., last 10 years or 5 years), etc. that were used the collect the related references to prepare this review manuscript.  

7.       Line 112: At the beginning on this section, state why pre-treatments of WPCBs is required (i.e., state the potential benefits). How it would impact the efficiency of the downstream recycling processes.

8.       Table 2: The heading of the column 4 needs to be changed from “NPs” to “Nanoparticles” or the abbreviation can be elaborated in the table footnote.

9.       Figure 2: State the source of the data. If collected from the literature, please cite the reference with proper format.

10.   Line 163: “4.2. Hydrometallurgical processes”. Based on the Table 3 data, add some discussion is encouraged about the variations of performance of the hydrometallurgical processed reported in various studies for the leaching/recovery of metals from WPCBs.  

11.   After the hydrometallurgical process, authors are suggested to include a section on the pyrometallurgical process by emphasizing the key operational steps as well as the process performance.

12.   For readers better understanding, Table 4 can be rearranged by initially providing the bacterial leaching followed by fungal bioleaching and biosorption. At present, bacterial and fungal bioleaching are mixed. The table title should be more specific, for example, the text “E-waste” should be replaced by “WPCBs”. Similarly, the title of Table 5 should be more specific instead of general term, like “E-waste”.

13.   Line 326 – 493: “6. Green Synthesis of Nanoparticles from WPCBs”. In this section, authors have reported various green methods for synthesis of nanoparticles from WPCBs. At the end of this section, it would be good to have a comparative discussion on the potential advantages and disadvantages of different methods.

14.   Line 601 – 641: The conclusions section is too long and not focused. It should be condensed with the major findings that the authors found with critical analysis of the literature.

15.   After conclusions, add a section on the “Future perspectives” by discussing the key knowledge gaps on the review topic and your recommendations for the future works which would help to develop efficient, cost-effective and environmentally benign WPCBs recycling processes.   

16.   The in-text reference should be in a uniform format (e.g., Dias et al., 2022).

17.   The paragraphing is not wisely done throughout the manuscript. In some places, only two lines are included in a paragraph. Thus, the authors need to pay a close attention to prepare a well-structured manuscript with logical flow of information from one paragraph to the next which would enhance the readability of the review.

Comments on the Quality of English Language

Minor English editing. 

Author Response

Thank you to the reviewer for your valuable feedback on the manuscript. Below, you will find point-by-point responses addressing each of your comments. Your input has been instrumental in enhancing the quality of the manuscript.

  1. Abstract: At the end of the abstract, add a statement on the important implication of this review.

R:/ The end of the abstract was reworded from the previous version, and the following statement was added in return: “The important implication of this review lies in its revelation of sustainable metal nanoparticle synthesis from WPCBs as a pivotal solution to E-waste environmental concerns, paving the way for eco-friendly recycling practices and the supply of valuable materials for diverse industrial applications.

  1. Line 46: “rare earth elements”. Could you list the dominant elements here.

R:/ The main elements found in the E-wastes were added.

  1. Line 52 – 56: Need reference for the statements.

R:/ The references for the statements of this paragraph were included ant the end, as suggested by the reviewer.

  1. In introduction, authors should strengthen their argument why it is necessary to recycle WPCBs from the perspectives of environmental benefits, resource recovery and circularity.

R:/ We added a paragraph to strengthen the argument about the necessity to recycle WPCB. Recycling WPCBs is imperative, offering a triad of compelling reasons rooted in environmental stewardship, resource conservation, and the promotion of circularity. From an environmental standpoint, the responsible recycling of WPCBs mitigates the hazardous impact associated with improper disposal, preventing the release of toxic elements into the environment. Embracing a circular economy, WPCB recycling promotes the reuse of extracted materials in manufacturing processes, reducing the reliance on virgin re-sources, and minimizing overall waste generation.

  1. In the last paragraph of the introduction, justification is sought on the novelty and importance of this review with respect to the existing published review articles. Why this review should be published and how it will contribute to the academics and industries who are working very hard for developments of novel WPCBs recycling methods for optimum recovery of valuable metals.

R:/ The end of the introduction section was modified to remark the importance and contributions of this study.

  1. In the introduction or as a separate section, develop a section on the “Review Method” by explaining the selection criteria such as the scientific database (Scopus, Web of Science and/or Google Scholar, etc.), the relevant keywords, the publication period (e.g., last 10 years or 5 years), etc. that were used the collect the related references to prepare this review manuscript.  

R:/ We thank the reviewer for this recommendation. However, we wish to point out that this work is not a systematic review of the literature, and that more than 93% of the bibliographic sources were selected from the last 10 years. We respectfully consider that it is inconvenient to burden the introduction with more information on the methodology, understanding that the way in which we conducted this work consists mainly of creating a list of relevant articles according to a set of keywords, reading them and extracting from them the most relevant information to be printed in the manuscript.

  1. Line 112: At the beginning on this section, state why pre-treatments of WPCBs is required (i.e., state the potential benefits). How it would impact the efficiency of the downstream recycling processes.

R:/ It is essential to employ any of these techniques, especially before the leaching process for enhancing the efficiency of dissolution and reduce the consumption of energy.

  1. Table 2: The heading of the column 4 needs to be changed from “NPs” to “Nanoparticles” or the abbreviation can be elaborated in the table footnote.

R:/ The heading of the column 4 was changed to “Nanoparticles”.

  1. Figure 2: State the source of the data. If collected from the literature, please cite the reference with proper format.

R:/ The references were added in Figure 2 to remark the source of information.

  1. Line 163: “4.2. Hydrometallurgical processes”. Based on the Table 3 data, add some discussion is encouraged about the variations of performance of the hydrometallurgical processed reported in various studies for the leaching/recovery of metals from WPCBs.  

R:/ A paragraph was added with the discussion of the data presented in Table 3 (updated as Table 4): where aqua regia is still the mixture of acids most used in hydrometallurgical processes to obtain Au, Ag and Pd from E-waste. The use of halogens in the extraction processes has shown obtaining an efficiency close to 100% for Au, however, the most recent studies have opted for the use of more environmentally friendly solvents but with a great effectiveness in as thiourea and thiosulfates.

  1. After the hydrometallurgical process, authors are suggested to include a section on the pyrometallurgical process by emphasizing the key operational steps as well as the process performance.

R:/ Thank you very much for the recommendation. However, we believe that the inclusion of pyrometallurgical processes deviates from the sustainable approach. While there are opportunities to improve the sustainability of pyrometallurgical processes, careful consideration needs to be given to energy efficiency, environmental impact and technological feasibility.

  1. For readers better understanding, Table 4 can be rearranged by initially providing the bacterial leaching followed by fungal bioleaching and biosorption. At present, bacterial and fungal bioleaching are mixed. The table title should be more specific, for example, the text “E-waste” should be replaced by “WPCBs”. Similarly, the title of Table 5 should be more specific instead of general term, like “E-waste”.

R:/ Table 4 (updated to Table 5) was originally organized according to the Green recovery methods. However, we have chosen to change the order, according to the recommendations of the evaluator by initially providing the bacterial leaching followed by fungal bioleaching and biosorption.

  1. Line 326 – 493: “6. Green Synthesis of Nanoparticles from WPCBs”. In this section, authors have reported various green methods for synthesis of nanoparticles from WPCBs. At the end of this section, it would be good to have a comparative discussion on the potential advantages and disadvantages of different methods.

R:/ A discussion was added at the end of the Green Synthesis of Nanoparticles from WPCBs, indicating the main advantages and limitations of each method.

Green synthesis is presented as a sustainable alternative for the valorization of WPCBs compared to traditional methods. Within the green techniques mentioned here, some have important aspects that are presented as advantages, for example, obtaining nanoparticles using organic acids reduces synthesis times, however, the stability of the nanoparticles obtained and their biocompatibility, in general, is reduced. On the other hand, the use of microorganisms, such as bacteria and fungi, provide nanoparticles with defined and stable formats, however, the production cycles are broader and there are limitations in the inability to operate at high pulp densities thus limiting the potential. for profitability. The use of plant extracts generates stable nanoparticles, but studies report that the proportion of the plant extract and chemical solution was the primary factor that affects the size of NPs and stability. Impurities of the extracts, limitations in process engineering, and operation stability are considered significant is-sues. Algae have demonstrated efficiency in the synthesis and stability of the nanoparticles generated, but they present limitations in the variability in the synthesis processes.

  1. Line 601 – 641: The conclusions section is too long and not focused. It should be condensed with the major findings that the authors found with critical analysis of the literature.

R:/ According to the feedback received by all the reviewers, the conclusions section was reformulated.

  1. After conclusions, add a section on the “Future perspectives” by discussing the key knowledge gaps on the review topic and your recommendations for the future works which would help to develop efficient, cost-effective and environmentally benign WPCBs recycling processes.   

R:/ We appreciate the reviewer suggestions. Along with the conclusions, future perspectives and some considerations were added.

  1. The in-text reference should be in a uniform format (e.g., Dias et al., 2022).

R:/ The format for in-text references was checked and adjusted as suggested by the reviewer

  1. The paragraphing is not wisely done throughout the manuscript. In some places, only two lines are included in a paragraph. Thus, the authors need to pay a close attention to prepare a well-structured manuscript with logical flow of information from one paragraph to the next which would enhance the readability of the review.

R:/ The authors checked the paragraphing, and some adjustments were included to improve the readability. We appreciate the reviewer's suggestions.

Reviewer 5 Report

Comments and Suggestions for Authors

The review is very interesting and up-to-date. However, some important issues should be clarified and corrected:

1. As alternative leaching agents many researchers indicate such compounds as ionic liquids or deep eutectic solvents. For example:
a. Waste Management 78 (2018) 191–197
b. Hydrometallurgy 205 (2021) 105730
c. J. AM. CHEM. SOC. 2004, 126, 9142-9147

2. Broad references [11-15] by for example: See also, for example:
- Science of the Total Environment 859 (2023) 160391
- Electronic wastes, Physical Sciences Reviews. 2018; 20180020

3. Lines 126-129 repeat exactly lines 130-133.

4. Lines 172-173 mention also alternative leaching agents, e.g. ionic liquids or deep eutectic solvents: 
a. Waste Management 78 (2018) 191–197
b. Hydrometallurgy 205 (2021) 105730
c. J. AM. CHEM. SOC. 2004, 126, 9142-9147

5. Issues to biosorption: What about further treatment of the metals sorbed on biomaterial? If metals/metal ions are to be recovered and reused they must be desorbed. And the problem is that metal ion delution results in dilution of the metal ions in the eluent, and then the recovery is very difficult. Comment on it.

6. To Table 5: 
a. To have nanoparticles, at first, they must be synthesized. Here the Authors do not mention methods of their synthesis and do not prove that they are present in the solutions which are treated with the methods shown in Table 5. For example, I've never heard that solvent extraction is used for nanoparticle separation.
b. Why table caption says about nanoparticles from e-waste but there is no connection with metal nanoparticles shown in none of the methods presented in the table? The caption should be changed.

7. Line 309: Give example of exact solution containing nanoparticles and show application of dialysis to separate them. What is the size of pores in dialysis membranes? Is it really possible to separate nanoparticles with such membranes? Nanoparticles of metals and metal ions must be clearly distinguished, this is not the same. 

8. Lines 317-324: But still there are no metal nanoparticles.

9. Line 357: What was the Ag concentration in the feed for production of nano silver? And how is it related to silver concentration in solutions after e-waste leaching?

10. In Table 6, column "Element": It would be good to give size of the nanometals and efficiency of nanometal synthesis.

11. Line 614: The presented methods are not used for nanoparticles separation, these could be used for concentration or separation of metal ions, and then after using these methods the solutions can be used as feeds for nanometal synthesis.

Author Response

Thank you to the reviewer for your valuable feedback on the manuscript. Below, you will find point-by-point responses addressing each of your comments. Your input has been instrumental in enhancing the quality of the manuscript.

As alternative leaching agents many researchers indicate such compounds as ionic liquids or deep eutectic solvents. For example:
a. Waste Management 78 (2018) 191–197
b. Hydrometallurgy 205 (2021) 105730
c. J. AM. CHEM. SOC. 2004, 126, 9142-9147

Lines 172-173 mention also alternative leaching agents, e.g. ionic liquids or deep eutectic solvents

R:/ Thank you for this review.

Broad references [11-15] by for example: See also, for example:
- Science of the Total Environment 859 (2023) 160391
- Electronic wastes, Physical Sciences Reviews. 2018; 20180020

Lines 126-129 repeat exactly lines 130-133.

R:/ Thank you for this review. The mistake was corrected.

Issues to biosorption: What about further treatment of the metals sorbed on biomaterial? If metals/metal ions are to be recovered and reused they must be desorbed. And the problem is that metal ion delution results in dilution of the metal ions in the eluent, and then the recovery is very difficult. Comment on it.

R:/ We added a comment concerning the way to recover metal ions after biosorption. We remarked the use of weak organic acids to recover adsorpted metals.

To Table 5: To have nanoparticles, at first, they must be synthesized. Here the Authors do not mention methods of their synthesis and do not prove that they are present in the solutions which are treated with the methods shown in Table 5. For example, I've never heard that solvent extraction is used for nanoparticle separation.

R:/ Dear reviewer, section 6 introduces selected green methods for nanoparticle synthesis. In the case of Table 5, the metal extraction, specifically gold from WPCBs is enhanced by the incorporation of Organophosphorus derivatives, guanidine derivations, and a mixture of amines−organophosphorus derivatives, prior to chemical reduction. We rewrite the sentence to clarify.  

To Table 5: Why table caption says about nanoparticles from e-waste but there is no connection with metal nanoparticles shown in none of the methods presented in the table? The caption should be changed.

R:/ The caption was changed to Concentration and purification of metals for nanoparticles production from WPCBs, to improve clarity for this table.

Line 309: Give example of exact solution containing nanoparticles and show application of dialysis to separate them. What is the size of pores in dialysis membranes? Is it really possible to separate nanoparticles with such membranes? Nanoparticles of metals and metal ions must be clearly distinguished, this is not the same. Lines 317-324: But still there are no metal nanoparticles.

R:/ The caption was changed to Concentration and purification of metals for nanoparticles production from WPCBs, to improve clarity for this table.

Line 357: What was the Ag concentration in the feed for production of nano silver? And how is it related to silver concentration in solutions after e-waste leaching?

R:/ The initial concentration in the feed of Ag was reported as 15.7 g.L−1. In this study, The solution containing silver took about 25 min to change from translucent (onset of dripping) to light yellow, and an additional 5 min of cold stirring was maintained until a strong yellow color was obtained. The ORP value dropped from 524 mV (initial - after the end of the citrate drip) to 467 mV (final - after cold stirring and complete reduction of silver) indicating a reduction in silver and formation of nanoparticles.

In Table 6, column "Element": It would be good to give size of the nanometals and efficiency of nanometal synthesis.

R:/ The size of the nanomaterials after nanometal synthesis was included in the Table, as suggested by the reviewer.

Line 614: The presented methods are not used for nanoparticles separation, these could be used for concentration or separation of metal ions, and then after using these methods the solutions can be used as feeds for nanometal synthesis.

R:/ The reviewer is right. The conclusions were rewritten for clarity, as suggested also by other reviewers.

Round 2

Reviewer 2 Report

Comments and Suggestions for Authors

The authors have made a detailed response to criticisms, most of which seem reasonable when considered with the revised manuscript. An attempt has been made to make it more critical, but issues remain. A number of responses do not address the issues raised as first detailed.

"Biosorption is said to be a passive physicochemical  process"lines 280/1, but this is illustrated in lines 302-305, by ion exchange reactions, which involve coordination of metal ions being trapped by functionalities on polysaccharides. This is not a  passive process but a chemical reaction. Then lines 305-7 list functional groups that can trap metals by coordination, again chemical bioabsorption. This is the point I made before. Passive biosorption is being illustrated by chemisorption examples. A passive process might be if the biological material simply filtered out metal salts.

Lines 419-421 and 429-433 still maintain that the production of CuO nanoparticles from a plant extract or from copper nitrate involve plant induced reduction processes. This was the point objected to in the first report. CuO is in the +II oxidation state, the highest observed for environmental Cu, so it cannot be formed by reduction. Use the correct chemical terms!

Original lines 438/9, the Qu was asked "what was in the clear colloidal broth for it to be relevant ? This was not answered and the section was changed lines 476 -9 to a statement that makes no sense.

Former lines 471-2 appear to be contradicted by three examples in line 462-470. Still in lines 518/9, this proposition is contradicted by examples given in preceding lines.

"Cu does not dissolve in water" A response was given to this "What the article means it is better dispersed" Here is the quotation from the paper.

2.3.2. Synthesis of mag-MOF(Cu)

First, 0.21 g Cu (recovered from WPCBs) was dissolved in 3 mL pure

water followed by the dissolution of 0.10 g 2-MIM in 3 mL ethanol.

These two solutions were combined and fully agitated for 20 min.

The reviewers have no basis for their reinterpretation "as better dispersed" as the article states clearly two solutions were mixed. moreover if it were  dispersion of Cu metal, the following chemistry would not occur. Further, the reviewers cannot reinterpret this without the authors' permission. As it stand the CuMOF formation is based on a nonsense and cannot be used as an illustration. Further, and no response was made to the point that the production of the CuMOF involved a succession of complex reagents, one of which has to be handled under nitrogen, making it an unattractive catalyst.. The role of recovered Cu is relatively minor.

Turning to the main text.

Inclusion of ref 14 is welcome, but more could have been made of the relative values of the various component of the WPCBs that are evaluated there.

Table 3 Toxicity of thiourea....It should be handled as a carcinogen, and is thus more toxic than thiosulfate. Line2 202-4 Cyanidation ......".produces excellent environmental" What does this mean? Cyanide readily oxidises to less toxic cyanate under environmental conditions or in iron rich conditions is converted into less toxic ferricyanide.

Lines 210/211 Make no sense ..reword. Table 4 What does iodine-iodine mean?

Table 4  Thiourea should be handled as a carcinogen and is thus much more toxic than thiosulfate and they should not be ranked the same. In addition, there will be disposal problems that should be highlighted for thiourea, as it has been done for thiosulfate.

Table 4 What does iodine-iodine mean?

Fig 4 Acidolysis equation is not balanced  M is in +II oxidation state in reagent and in n+ in the product implying some redox has occurred, The number of Hs is different on each side of the equation

Table 5 Three biosorption entries make no sense, those for Au, Cu, La. If these are to be removed from WPCBs with the first two present in metallic form, it requires a chemical oxidation reaction, which is not biosorption. Likewise even if La is present as the oxide, it requires an acidolysis reaction  to dissolve it and this is not biosorption

Tanle 6 The sequence of the first entry must be reversed . If one starts at pH 11-12,  three metals will be precipitated together and  only Zn will remain in solution. How are the ions precipitated ? presumably as hydroxides? This should be stated. How is zinc removed as it will be in solution in ammonia not precipitated. Mg(OH)2 solution?  It is largely insoluble in water. The process will produce a Ca,Mg, Fe, NH3 waste stream.

Line 426  S/L explain.  Lines 477-9 make no sense, reword   Line 486 delete “it”

Table 7 -COOH is not reducing  except under extreme thermal conditions

-C6H5OH should be -C6H4OH

Lines 551-3 reword, currently unintelligible Line 560, new sentence needed at “among”

Line 562 what is meant by “said waste”

Lines 673-5 and 682-3 are partly duplicated and could be combined

Lines 697-9 should be three sentences not one, and omit ”on the other hand”

 Line 703, what evidence has been presented for high selectivity of biosorption, and if there is cross reference it here.

Lines 722-5 should be two sentences

The references are much improved, but 124 and 127 have errors

Comments on the Quality of English Language

Generally OK, but new section in red are unsatisfactory from viewpoint of sentence construction. Reference abbreviations require full stops

Author Response

Dear reviewer, we hereby respond to all of your comments. You will find in this document the consolidated point-by-point responses. The authors thank you for your thorough review and the contributions received. Thank you very much.

"Biosorption is said to be a passive physicochemical process"lines 280/1, but this is illustrated in lines 302-305, by ion exchange reactions, which involve coordination of metal ions being trapped by functionalities on polysaccharides. This is not a passive process but a chemical reaction. Then lines 305-7 list functional groups that can trap metals by coordination, again chemical bioabsorption. This is the point I made before. Passive biosorption is being illustrated by chemisorption examples. A passive process might be if the biological material simply filtered out metal salts.

R:/ The biosorption process involves chemisorption, complexation, adsorption-complexation on the surface and in pores, ion exchange, microprecipitation, condensation of heavy metal hydroxides and surface adsorption, as presented by https://doi.org/10.1016/j.chemosphere.2022.135957.The word "passive" was omitted in the final version.

Lines 419-421 and 429-433 still maintain that the production of CuO nanoparticles from a plant extract or from copper nitrate involve plant induced reduction processes. This was the point objected to in the first report. CuO is in the +II oxidation state, the highest observed for environmental Cu, so it cannot be formed by reduction. Use the correct chemical terms!

R:/ For this aspect we have reviewed the original literature and actually what happens are oxide- reduction reactions where the metal species is reduced to a zero-oxidation state, followed by an oxidation process that forms a stable oxide, in this case CuO. These adjustments were incorporated in the document, adding the word oxide-reduction process.

Original lines 438/9, the Qu was asked "what was in the clear colloidal broth for it to be relevant ? This was not answered and the section was changed lines 476 -9 to a statement that makes no sense.

R:/ This question was resolved on line 497-500 of the new version of the manuscript as follows: "In the extracellular synthesis routes, fungal proteins, enzymes, cofactors, and metabolites like organic acids (e.g., citric acid, oxalic acid) play vital roles in the organism's survival and contribute to the reduction of metal ions into nanoparticulate forms." Additionally, the last paragraph on lines 477-480 of the new version of the manuscript "Finally, the management of waste generated in the plant leaf extract is mainly grouped into: Thermochemical treatments as combustion, gasification, and pyrolysis; bi-ochemical treatments to obtained, for example bioethanol; drying methods; condensation of active components" was suggested by another reviewer in the previous round.

Former lines 471-2 appear to be contradicted by three examples in line 462-470. Still in lines 518/9, this proposition is contradicted by examples given in preceding lines.

R:/ In response to this comment, the authors left in this new version only the suggestion for new works that could be developed on this topic, and decided to eliminate the line that indicated the presence of few works on the topic.

"Cu does not dissolve in water" A response was given to this "What the article means it is better dispersed" Here is the quotation from the paper.

2.3.2. Synthesis of mag-MOF(Cu)

First, 0.21 g Cu (recovered from WPCBs) was dissolved in 3 mL pure

water followed by the dissolution of 0.10 g 2-MIM in 3 mL ethanol. These two solutions were combined and fully agitated for 20 min.

The reviewers have no basis for their reinterpretation "as better dispersed" as the article states clearly two solutions were mixed. moreover if it were dispersion of Cu metal, the following chemistry would not occur. Further, the reviewers cannot reinterpret this without the authors' permission. As it stand the CuMOF formation is based on a nonsense and cannot be used as an illustration. Further, and no response was made to the point that the production of the CuMOF involved a succession of complex reagents, one of which has to be handled under nitrogen, making it an unattractive catalyst.. The role of recovered Cu is relatively minor.

R:/ When reviewing in detail the article in mention https://doi.org/10.1016/j.jenvman.2023.118755, we found that indeed the methodology proposed by the original authors contains a technical inaccuracy when published, by erroneously indicating that copper is dissolved, in this sentence: "First, 0.21 g Cu (recovered from WPCBs) was dissolved in 3 mL pure", however, the only thing we do in this review was to cite the novel results of the article as a pesticide degradation agent.

Inclusion of ref 14 is welcome, but more could have been made of the relative values of the various component of the WPCBs that are evaluated there.

R:/ We appreciate the evaluator's suggestion, however, we felt that this would make the discussion longer and we were actually responding to another suggestion received by another evaluator.

Table 3 Toxicity of thiourea....It should be handled as a carcinogen, and is thus more toxic than thiosulfate. Line2 202-4 Cyanidation  ".produces excellent environmental" What does this mean? Cyanide readily oxidises to less toxic cyanate under environmental conditions or in iron rich conditions is converted into less toxic ferricyanide.

R:/ We have taken on board the recommendations received and have modified these aspects in Table 3.

Lines 210/211 Make no sense ..reword. Table 4 What does iodine-iodine mean?

R:/ The correct term is "Iodine-Iodide", which are two species derived from iodine. This was a typo and has been corrected.

Additional comments

 Fig 4 Acidolysis equation is not balanced M is in +II oxidation state in reagent and in n+ in the product implying some redox has occurred, The number of Hs is different on each side of the equation

R:/ In the acidolysis equation, the oxidation state of M is not necessarily +II. For this reason, we write it generically as MO (Metal Oxide). That "O" is not the oxygen atom, but the acronym for the word "Oxide", as it appears in the lower box of the Figure 4. However, we revised the balancing and chose to adjust the coefficients of the acidolysis equation a little to improve the clarity of the stoichiometry with respect to hydrogen. Thanks to the reviewer for bringing this to our attention.

Table 5 Three biosorption entries make no sense, those for Au, Cu, La. If these are to be removed from WPCBs with the first two present in metallic form, it requires a chemical oxidation reaction, which is not biosorption. Likewise even if La is present as the oxide, it requires an acidolysis reaction to dissolve it and this is not biosorption

R:/ To resolve these concerns of the reviewer, we have verified the original source, given that we started from a study that we were not part of, and contrasted it with other studies that confirm that what is stated in the Table is technically correct. A more detailed explanation follows:

Regarding Au: Hybrid combination. Several reactions like acidolysis, complexolysis, and redoxolysis can take place when microbes are placed in contact either with solid metal-containing matrix or metals in solution (Ledin and Pedersen, 1996). Bacterial interaction with Au3+occur through O and N containing active groups of proteins and cell wall like hydroxyl of saccharides, amide group in peptide bond and the carboxylate anion containing amino-acid residues in polypeptide backbone. In aqueous acidic environment, biomass surface is positively charged because of the protonation of these O and N active groups. It has been experimentally determined that release of the chloride ion from adsorbed AuCl4 and adsorption of gold both occur concurrently. The reaction proceeds via the formation of ion pairs between oppositely charged AuCl4 and protonated active groups; thus eliminating an ion pair of chloride (Greene et al., 1986). AuCl4 interacts with biomass by rapidly reducing Au3+ to Au+ followed by slow reduction to Au0.

La and Tb: what actually occurs is bioaccumulation, where cells actively encapsulate and accumulate the metal both La and Tb. Concerning metals bioaccumulation techniques, Qu and Lian (2013) showed the advantages deriving from the employment of heterotrophic microorganisms, such as filamentous fungi. Fungi are able to develop metal tolerance and resistance mechanisms, such as metallothionein or phytochelatin proteins, which may bind and deactivate toxic metals, or enable the heavy metal storage in vacuoles (Gadd, 2007, Onofri et al., 2011).

Cu and Fe: both processes occur, in this case, Pleurotus florida and Pseudomonas spp. were employed for the biosorption and bioleaching of copper and iron. This can be attributed to bio- catalysis by the laccase enzyme.

Table 6 The sequence of the first entry must be reversed . If one starts at pH 11-12, three metals will be precipitated together and only Zn will remain in solution. How are the ions precipitated ? presumably as hydroxides? This should be stated. How is zinc removed as it will be in solution in ammonia not precipitated. Mg(OH)2 solution? It is largely insoluble in water. The process will produce a Ca,Mg, Fe, NH3 waste stream.

R:/ Cu was adjusted to a pH range of 11–12 using Ca(OH)2. Zn was set at a pH of 11 with an ammonia solution. Pb was controlled within a pH range of 7–11 using a Mg(OH)2 solution, and Ni was maintained at a pH of 5.5 with Fe(OH)2. pH adjustments were made using 1.0 M HCl and 0.1 M NaOH.

The article does not specifically state in what form the metals are precipitated, most likely in the form of hydroxides. The methodology does not detail the preparation of Mg(OH)2, however, even though it is low, it is large enough to partially dissolve to produce ions in solution and thus presumably would allow precipitation of Pb. And of course, the process will produce a Ca, Mg, Fe, NH3 waste stream.

Line 426 S/L explain. Lines 477-9 make no sense, reword Line 486 delete “it”

R:/ (Line 426) The term S/L stands for Solid/Liquid proportion. This word was changed in the document. (Line 477-9) These lines were suggested by one of the reviewers, and they explain how is the the management of waste generated in the plant leaf extract. (Line 486) the “it” was erased. Thank you.

Table 7 -COOH is not reducing except under extreme thermal conditions

-C6H5OH should be -C6H4OH

R:/ The radical number (-C6H4OH) was changed correctly.

In the case of -COOH as reducing agent, in the original source of information Pekkoh et al. (2023) concludes that hydroxyl (−OH), carbonyl (C=O), amino (−NH2), carboxyl (−COOH), and phenolic in

Sargassum spp. can potentially serve as both stabilizing and reducing agents in the rapid biosynthesis of Ag/AgCl-NPs-ME through microwave-assisted synthesis.

Lines 551-3 reword, currently unintelligible Line 560, new sentence needed at “among”

R:/ A dot was missing in line 560 after “among”. It was included to separate the first idea to the second, related to the biological processes. Thank you for the correction.

Line 562 what is meant by “said waste”

R:/ The sentences was unclear and rewritten as: “(…) where added value can be given to the waste generated in green metal extraction processes”

Lines 673-5 and 682-3 are partly duplicated and could be combined

R:/ The duplicated information was combined to improve readability. Thank you for this suggestion.

Lines 697-9 should be three sentences not one, and omit ”on the other hand”

R:/ The writing of this part of the text was reviewed and the term “on the other hand” was removed.

Line 703, what evidence has been presented for high selectivity of biosorption, and if there is cross reference it here.

R:/ The provided reference from Gupta et al. (2019) https://doi.org/10.1016/j.molliq.2018.10.134 in section 9 describes in detail and evidences the selectivity of biosorbents.

Lines 722-5 should be two sentences

R:/ The writing of Lines 722-5, updated as lines 725-9 were reviewed and separated from the long paragraph in two sentences as suggested.

The references are much improved, but 124 and 127 have errors

R:/ References 124 and 127 were reviewed to avoid any mistake. Thank you.

Reviewer 3 Report

Comments and Suggestions for Authors

N.A.

Author Response

Dear reviewer, the authors thank you for your valuable feedback on the manuscript and we are glad to have had your concept evaluation.

Sincerely,

The Authors.

Reviewer 4 Report

Comments and Suggestions for Authors

The quality and readability of the manuscript is improved after revision. The revised version can be accepted. 

Comments on the Quality of English Language

Minor Editing. 

Author Response

(The authors gave the same response as above.)

Round 3

Reviewer 2 Report

Comments and Suggestions for Authors

Although the authors have improved the review, my concern remains that the authors are unwilling to challenge reported statements that are chemically incorrect. Just because authors state something does not make it true. . Reviews should critically examine the literature they are citing and where it is incorrect or does not make sense, they should say so. This is particularly the case with the "fractional chemical precipitation" procedure of Table 6. As I have already pointed out, adjusting the pH to 11-12 as apparently the first step will precipitate  Cu, Pb and Ni together, as any text on qualitative inorganic analysis will indicate. "Zinc was set at a pH of 11 with an ammonia solution" What does this achieve? Is this seeking to mask Zn from precipitation? But Zn does not precipitate at this pH , however the adjustment is made.. "Pb was controlled within a pH range of 7-11 with a Mg(OH)2 solution." To what end? etc. This is not a description of selective precipitation whatever the original authors state, and, as described, is a chemical nonsense. This is what I meant by lack of a critical approach to the literature.

Similarly just because Pekkoh et al state that carbonyl and carboxyl groups can act as reducing agents does not make it correct. Standard organic texts have carbonyl and carboxylic acids as the final oxidation products in oxidation processes unless one uses very  elevated temperatures where any organic substrate can be oxidised into CO2 and water. The reviewers are not obliged to restate  chemical implausibility even if in the literature.

The same applies to the CuMOF destruction of  pesticides. Are the authors so short of examples that they wish to persist with an example based on a chemical nonsense in the preparation? At least the authors should point out there is  a fault in the preparation of the CuMOF.

Comments on the Quality of English Language

There are issues of grammar and meaning that remain. New text brings new problems. For example the last sentence in the added lines 477-480, there are two errors, which affect the readability.

Author Response

Dear reviewer, we truly appreciate the thoroughness and expertise with which you approached the review process. Your comments and suggestions have been instrumental in refining our manuscript, and we believe that your input will greatly strengthen the overall quality of the paper. Here are the answers to your comments point by point.

Although the authors have improved the review, my concern remains that the authors are unwilling to challenge reported statements that are chemically incorrect. Just because authors state something does not make it true.. Reviews should critically examine the literature they are citing and where it is incorrect or does not make sense, they should say so. This is particularly the case with the "fractional chemical precipitation" procedure of Table 6. As I have already pointed out, adjusting the pH to 11-12 as apparently the first step will precipitate  Cu, Pb and Ni together, as any text on qualitative inorganic analysis will indicate. "Zinc was set at a pH of 11 with an ammonia solution" What does this achieve? Is this seeking to mask Zn from precipitation? But Zn does not precipitate at this pH , however the adjustment is made.. "Pb was controlled within a pH range of 7-11 with a Mg(OH)2 solution." To what end? etc. This is not a description of selective precipitation whatever the original authors state, and, as described, is a chemical nonsense. This is what I meant by lack of a critical approach to the literature.

R/: In Table 6, the reference was changed to another https://doi.org/10.1016/j.matpr.2023.08.199

Similarly just because Pekkoh et al state that carbonyl and carboxyl groups can act as reducing agents does not make it correct. Standard organic texts have carbonyl and carboxylic acids as the final oxidation products in oxidation processes unless one uses very  elevated temperatures where any organic substrate can be oxidised into CO2 and water. The reviewers are not obliged to restate chemical implausibility even if in the literature.

R/: This phenomenon occurs and requires certain operational conditions; however, they can act as reducing agents. According to a study by Zahoor et al. (“A review on silver nanoparticles: classification, various methods of synthesis, and their potential roles in biomedical applications and water treatment”), the bio-based synthesis of Ag-NPs from plant extracts suggests that secondary metabolites in these extracts can reduce Ag+. These secondary metabolites, such as vitamins, polysaccharides, amino acids, proteins, enzymes, polyphenolics, flavonoids, etc., play a crucial role as reducing agents for Ag+ and as size control agents in the formulation of Ag-NPs. Moreover, functional groups such as the carboxyl group (COOH) in glutamic and aspartic acid residues and the hydroxyl group (OH) of tyrosine residues have been discovered as key in maintaining the stability of Ag+ and the production of small, polydisperse Ag nanoplatelets. This supports the results of Mohamad et al. (“Plant extract as reducing agent in synthesis of metallic nanoparticles: A review”), who reported that the hydroxyl and carboxyl groups present could act as reducing agents and stabilizers in the synthesis of NPs. Conversely, bioactive compounds such as phenolics and alkaloids play a role in capping and stabilizing the produced NPs. These can be found here: https://doi.org/10.3390/nano13071244

In another study, they explain the reducing effect of carboxylates from Adipic acid on silver ions.

“By the action of reducing agent NaBH4, nucleation occurred, and silver ions/adipic acid complexes turned into AgNPs. Adipic acid is a dicarboxylic acid that possesses the ability to resonate a lone pair of electrons on oxygen with the carbonyl. It is well established that the carboxylate group (—COO−) is a functional group that is commonly used to stabilize silver nanoparticles [45,46].

The carboxylate moieties of adipic acid were adsorbed on the silver surface of AgNPs@AA with a bi-dentate bridging arrangement. The —COO− moieties are bound to silver (sp orbital) via two oxygen atoms of the carboxylate moieties”. Source: https://doi.org/10.3390/molecules27113363

From our experience, we have used different types of organic reducers such as citric acid, glucose, and sodium citrate among others with temperatures between 70-80°C with good results in the reduction of metallic nanoparticles as Silver.

The same applies to the CuMOF destruction of  pesticides. Are the authors so short of examples that they wish to persist with an example based on a chemical nonsense in the preparation? At least the authors should point out there is  a fault in the preparation of the CuMOF.

 R/: The example is related to applications derived from WPCBS waste, nanoparticles have multiple applications, but we find it interesting that this example is from WPCBS waste nanoparticles and is innovative.

There are issues of grammar and meaning that remain. New text brings new problems. For example the last sentence in the added lines 477-480, there are two errors, which affect the readability.

R/: The added text in lines 677 – 680 was grammar-checked as suggested by the reviewer.

Thank you.
